# Structure-Preserving Learning Improves Geometry Generalization in Neural PDEs

**Benjamin D. Shaffer** [1]   **Shawn Koohy** [1]   **Brooks Kinch** [1]   **M. Ani Hsieh** [1]   **Nathaniel Trask** [1 2]

## Abstract

We aim to develop *physics foundation models* for science and engineering that provide real-time solutions to Partial Differential Equations (PDEs) while preserving structure and accuracy on unseen geometries. To this end, we introduce *General-Geometry Neural Whitney Forms* (Geo-NeW): a data-driven finite element method. We jointly learn a differential operator and compatible reduced finite element spaces defined on the underlying geometry. The resulting model is solved to generate predictions, while exactly preserving physical conservation laws through Finite Element Exterior Calculus. Geometry enters the model as a discretized mesh both through a transformer-based encoding and as the basis for the learned finite element spaces. This explicitly connects the underlying geometry and imposed boundary conditions to the solution, providing a powerful inductive bias for learning neural PDEs, which we demonstrate improves generalization to unseen domains. We provide a novel parameterization of the constitutive model ensuring the existence and uniqueness of the solution. Our approach demonstrates state-of-the-art performance on several steady-state PDE benchmarks, and provides a significant improvement over conventional baselines on out-of-distribution geometries. Code is provided here.

## 1 Introduction

*Scientific foundation models* aim to reduce the cost of solving partial differential equations (PDEs) by learning reusable surrogate models across physical systems, geome-

tries, and parameter regimes (Herde et al., 2024; McCabe et al., 2025; Shen et al., 2024; Choi et al., 2025; National Academies of Sciences & Medicine, 2024). Unlike large language models trained on large corpora of text, scientific data are unstructured, sparse, and expensive to curate, making the large-scale corpus construction required for foundation models a major challenge. In domains where high volumes of data exist, such as weather prediction (Bodnar et al., 2024; Price et al., 2025) or protein/molecular physics (Jumper et al., 2021; Ahmad et al., 2022), learned simulators enable real-time predictions on single GPUs, unlocking design exploration previously restricted to supercomputing campaigns.

Recent work aims to curate corpora of PDE simulations for diverse fundamental physics over ranges of boundary conditions and material parameters (Ohana et al., 2024; Takamoto et al., 2022). These efforts typically restrict attention to rectilinear domains (or domains diffeomorphic to them), allowing the use of Cartesian representations. Complex geometry is a central unresolved barrier to deploying scientific foundation models in realistic engineering settings. In this work, we take geometry generalization to be a necessary capability for scientific foundation models intended for engineering design and simulation.

Most methods for geometric operator learning are posed as a regression from geometry and boundary conditions to a solution. Inspired by the Dirichlet-to-Neumann map from PDE theory, which links boundary geometry to solution operators (Lassas et al., 2003), we instead pose the learning problem as identifying a reduced-order finite element model which links coordinate-free representations of both physics and geometry through strongly imposed boundary conditions. The finite element model uses two geometry-conditioned neural fields to define: **1.** a reduced finite element space and **2.** an integral balance law on that space. Finite element exterior calculus (FEEC) ensures topological structure is respected with guarantees of stability, boundary condition enforcement, and discrete conservation. Geometry conditioning combines coordinate-free descriptors (heat kernel signature, harmonic coordinates) with metric information (signed-distance fields) via transformer encoding. At inference, the framework produces a reduced conventional finite element system solvable in real time on new meshes.

---

[1]Department of Mechanical Engineering and Applied Mechanics, University of Pennsylvania, Philadelphia, Pennsylvania, United States of America [2]Sandia National Laboratories, Albuquerque, New Mexico, United States of America. Correspondence to: Benjamin Shaffer <ben31@seas.upenn.edu>.

*Proceedings of the $43^{rd}$ International Conference on Machine Learning*, Seoul, South Korea. PMLR 306, 2026. Copyright 2026 by the author(s).

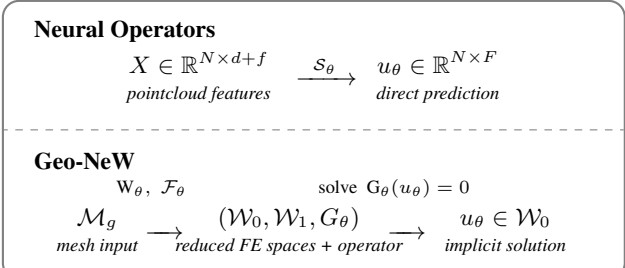

Schematic comparison of neural operators and Geo-NeW, we learn a physical model which provides a strong connection to geometry and useful inductive bias for generalization.

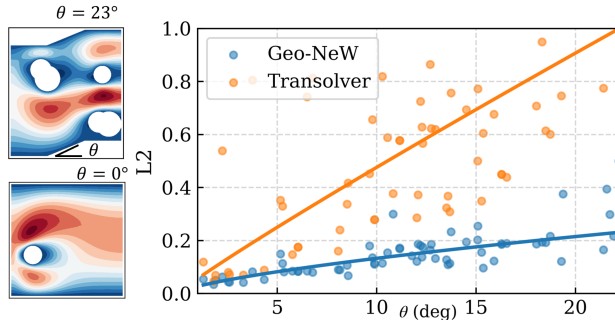

*Figure 1.* Geo-NeW provides improved generalization capability for strongly out-of-distribution geometries. Here, models trained on square domains with randomly circular obstacles, but evaluated on a domain including a variable angle step $\theta$; increasing $\theta$ provides a continuous measure of departure from the training distribution. We demonstrate reduced error compared to Transolver, which fails to produce meaningful predictions beyond $\theta = 20°$.

**Primary contributions: 1.** The first model that enforces boundary conditions and conservation exactly while generalizing across geometry. **2.** An extension of the FEEC framework developed in (Kinch et al., 2025; Actor et al., 2024; Trask et al., 2022) that enables learning physics on distinct geometries without retraining. **3.** A combination of geometry-aware encodings (HKS, HC, SDF) together with a corresponding transformer architecture that yields translation and rotation-invariant geometric embeddings and a discretization-invariant prescription of partitions. **4.** A physics parameterization that is convex in the solution variables but not in the geometric variables, which allows us to prove the existence and solvability of the learned models.

In engineering and scientific design, finite element simulations have value beyond predicting solutions of PDEs; they support multiphysics coupling, uncertainty quantification, sensitivity analysis, and eigenvalue estimation for stability studies. We match state-of-the-art performance on in-distribution operator learning tasks while demonstrating that physics-based structure provides a strong inductive bias for out-of-distribution generalization (See Fig. 1). Our approach also outperforms unconstrained transformer baselines on in-distribution inference for several tasks, despite the reduced capacity due to the imposed physical structure.

### 1.1 Related Work.

**Operator learning.** Operator learning has rapidly progressed from integral-kernel and Fourier approaches to include transformer-based architectures capable of handling large datasets and heterogeneous PDE families (Wang et al., 2025; Hao et al., 2023; Li et al., 2023c; Alkin et al., 2024). Mesh-based surrogates are also widely used, including graph-based methods (Janny et al., 2023; Pfaff et al., 2020; Brandstetter et al., 2022b) and mesh-informed operator learning in finite element spaces (Franco et al., 2023). Many scalable PDE surrogates follow an *encoder–approximator–decoder* (Seidman et al., 2022) paradigm: high-resolution fields are compressed into a small set of latent tokens, processed with global attention in latent space, and decoded back to the mesh (Wen et al., 2025; Alkin et al., 2024; Wu et al., 2024). Recent methods instantiate this compression

via inducing-point supernodes (Alkin et al., 2024; 2025; Lee & Oh, 2024) or slice-based tokenization (Wu et al., 2024; Luo et al., 2025; Hu et al., 2025; Tiwari et al., 2025), enabling efficient learning on large unstructured domains. In contrast, our latent variables are physically interpretable coefficients of a geometry-conditioned reduced finite element space, and the *approximator* corresponds to solving a network-defined conservation law through an implicit PDE solve rather than a transformer update step (Kinch et al., 2025).

**Implicit-layers and differentiable-optimization.** Our training procedure uses implicit differentiation through a nonlinear solver, and is related to implicit layers and differentiable optimization (Amos & Kolter, 2017; Agrawal et al., 2019). However, our equilibrium condition corresponds to a discretized *physical operator* derived from conservation laws and finite element structure, not a latent representation layer. As a result, the equilibrium represents a PDE solution and inherits conservation/stability structure from the discretization. Implicit models have been previously explored using FNOs (You et al., 2022) but without the finite element framework or structure preservation we present.

**Geometry generalization.** We categorize operator learning across geometric domains into three categories. (i) *Diffeomorphisms* or reference-domain mappings warp irregular geometries to a canonical coordinate system and then apply grid-based neural operators (Li et al., 2023a; Yin et al., 2024). (ii) *Graph-based models* operate directly on unstructured discretizations using message passing, mesh kernels, or neural operators (Sanchez-Gonzalez et al., 2020; Li et al., 2020a;b). (iii) *Point-cloud and transformer-based surrogates* avoid explicit grid representations by processing sets of spatial tokens, including transformer neural operators (Li et al., 2022), point-cloud neural operators such as GNOT (Hao et al., 2023) and Transolver (Wu et al., 2024), and latent-token operator-learning frameworks (Alkin et al.,

2024; Lee & Oh, 2024). Collectively, these approaches enable operator learning on irregular geometries by treating geometry as an input modality (via coordinate warping, graph connectivity, or point-cloud tokens) (Li et al., 2023a; Sanchez-Gonzalez et al., 2020; Hao et al., 2023; Wu et al., 2024; Alkin et al., 2024). In contrast, our approach treats the computational mesh as a tool for constructing mass and stiffness matrices with finite element spaces and physics prescribed by transformers.

**Structure-preserving machine learning.** This work requires notions of structure preservation drawn from both physics and geometric representation learning. Physics-informed architectures incorporate physical principles into neural models through soft (Lagaris et al., 1998; Raissi et al., 2017) or hard constraints. While many approaches target geometric structure in dynamical systems (for example, bracket or variational formulations) (Greydanus et al., 2019; Cohen & Welling, 2016; Brandstetter et al., 2022a; Chen et al., 2021), a substantial body of work also focuses on the structure of PDEs. In conventional PDE discretization, FEEC (and specifically Whitney forms) were developed to ensure that classical simulators inherit the topological structure of their continuum models (Arnold, 2018; Bochev & Hyman, 2006), particularly in electromagnetism (Bossavit, 1988). Our work builds on recent efforts (Kinch et al., 2025; Actor et al., 2024; Jiang et al., 2024) that construct neural architectures grounded in FEEC to produce data-driven models with the same solvability guarantees as standard finite element simulators. Recent geometric deep learning approaches (Bronstein et al., 2017) share the same mathematical/topological foundation as FEEC.

The physical structure governing conservation and stability is distinct from geometric representation learning that aims to preserve invariances such as $SO(3)/SE(3)$ symmetry or discretization invariance (Gerken et al., 2023; Thomas et al., 2018; Batzner et al., 2022). The coordinate-free graph embeddings used in our framework to obtain discretization-invariant neural fields are informed by contemporary techniques in geometric representation learning (Sun et al., 2009; Joshi et al., 2007; Ströter et al., 2024; Peng et al., 2022) using generalized coordinates to describe geometries.

## 2 Problem Setup

Let $g \in \mathcal{G}$ denote a geometry specification, including domain shape, topology, and boundary partitioning, and $\Omega_g \subset \mathbb{R}^d$ be the corresponding domain with boundary $\partial\Omega_g$. We consider steady-state physical systems governed by a PDE operator $\mathcal{P}$ and boundary operator $\mathcal{B}$ of the form

$$\underbrace{\mathcal{P}(u,\mu) - f}_{G(u,\mu,f)} = 0, \text{ in } \Omega_g \subset \mathbb{R}^d \quad (1)$$

$$\mathcal{B}(u) = u_b \text{ on } \partial\Omega_g \quad (2)$$

where $u : \Omega_g \to \mathbb{R}^F$ is the state of a conserved physical field, $f : \Omega_g \to \mathbb{R}^F$ is an external forcing, $u_b : \partial\Omega_g \to \mathbb{R}^F$ is boundary data, and $\mu : \Omega_g \to \mathbb{R}^k$ are field-valued physical parameters (e.g., material properties); for simplicity, we assume scalar or vector-valued parameters (e.g., Reynolds number) are treated as constant global features in $\mu$. The solution operator maps $u = \mathcal{S}(\alpha)$ for parameters $\alpha = \{g, \mu, f, u_b\}$. The aim of neural PDE learning is to identify $\mathcal{S}$ from data in the form of solution realizations $u^{(i)} = \mathcal{S}(\alpha^{(i)})$ for $i = 1, \ldots, N_{samples}$.

The geometry-dependent operator learning task is to predict $u$ for *unseen* $g$ for given $f$, $u_b$, and $\mu$. For each $g$, we assume access to a valid finite element mesh $\mathcal{M}_g$.

### 2.1 Implicit Operator Learning.

In contrast to typical operator learning approaches (e.g. Lu et al. (2021); Kovachki et al. (2023)) which would aim to regress $u = \mathcal{S}_\theta(\alpha)$, we instead construct approximate operators $G_\theta(u,\alpha) \sim G(u,\alpha)$ and $B_\theta(u,\alpha) \sim B(u,\alpha)$ such that $\mathcal{S}$ is defined implicitly:

$$\text{Find } u' \text{ such that } \quad G_\theta(u',\alpha) = 0, \quad \mathcal{B}_\theta(u') = u_b. \quad (3)$$

In this formulation, the learned object is the governing PDE operator rather than the solution map, and prediction requires solving a nonlinear system. We will construct $\mathcal{G}_\theta$ by designing **Component 1.** reduced order finite element spaces and **Component 2.** a coordinate-free representation of discrete physics so that the solution operator enforces boundary conditions, regularity, and conservation under extrapolation while guaranteeing efficient and stable solutions. These two components contain a conditioning mechanism that allows their associated transformers to condition on a latent encoding of the geometry (**Component 3**). In concert, this architecture produces a compact, easily solvable finite element space with a reduced description of physics that generalizes across unseen geometries.

### 2.2 Component 1: Reduced Finite Element Spaces

Whitney forms are finite element spaces which provide piecewise linear interpolants of differential forms ($k = 0$ nodal evaluations, $k = 1$ circulations on edges, $k = 2$ face fluxes, and $k = 3$ volumetric moments). These spaces preserve discrete div/grad/curl operations that guarantee the conservation structure inherent in the generalized Stokes theorems is preserved at a discrete level. Crucially, they support an exterior calculus which describes physics in a coordinate-free manner. For a given $g$, we construct reduced finite element spaces $\mathcal{W}_g^k$ as subspaces of low-order Whitney forms $\mathbf{W}_g^k$. When designing this dimension reduction, the so-called *exact sequence* property of Whitney forms (i.e., for a coboundary $d_k$, $d_k \mathcal{W}_g^k \subset \mathcal{W}_g^{k+1}$) must be preserved to ensure conservation structure and numerical stability are preserved at the reduced-order level (Actor et al., 2024). Denoting the lowest order basis $\mathbf{W}_g^0 = \text{span}(\phi_i(x))$. We

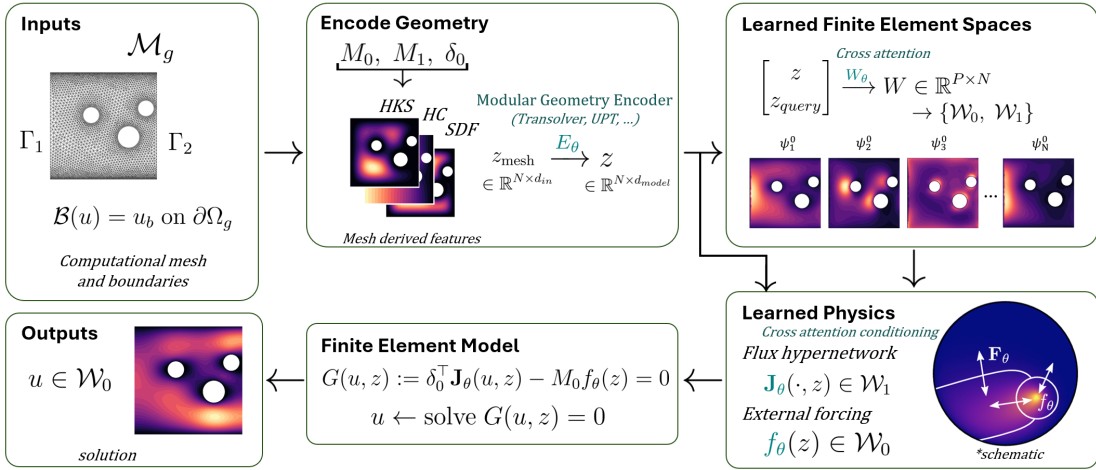

*Figure 2.* Geo-NeW pipeline for geometry-generalizable PDE surrogate modeling. A geometry encoder maps mesh-derived features to a latent context $z$ that conditions both reduced finite element spaces and a nonlinear flux model. Geometry enters both through the learned encoding and explicitly as a basis for the finite element model. The resulting learned finite element system explicitly encodes mesh topology and metric structure, enforces conservation and boundary conditions, and is solved implicitly to produce the PDE solution. Teal indicates a learnable component.

construct a conditional neural field $W(x, z)$ that assigns each basis function to a reduced basis, conditioned on an arbitrary vector $z$

$$\psi_i^0(x) = \sum_{j=1}^{N} W_i(x_j, z)\, \phi_j(x), \tag{4}$$

where $N = \dim(\mathbf{W_g^0})$, $W_i(x, z) \geq 0$ and $\sum_{i=1}^{P} W_i(x, z) = 1$ for all $j$, ensuring that $\sum_i \psi_i^0(x) = 1$ so fields admit interpretation as $P$ overlapping control volumes. The next order space is defined as

$$\mathcal{W}_g^1 = \text{span}\left(\psi_{ij}^1(x)\right)_{ij} = \left\{\psi_i^0 \nabla \psi_j^0 - \psi_j^0 \nabla \psi_i^0\right\}_{ij},$$

from which we easily show $\nabla \mathcal{W}_g^0 \subseteq \mathcal{W}_g^1$ therefore the exact sequence property is preserved. See Appendix B.

These spaces yield linear algebra building blocks sufficient for discretizing $H(div)$ conservation laws: the $L^2$-inner product induces mass matrices and the adjacency matrix $\delta$ encoding connectivity between adjacent partitions

$$M_0 = (\psi_i^0, \psi_j^0) \quad M_1 = (\psi_{ij}^1, \psi_{kl}^1), \quad (\psi_{ij}^1 \nabla \psi_k^0) = M_1 \delta. \tag{5}$$

We will use these blocks to build a mixed finite element discretization from which to learn dynamics.

### 2.3 Component 2: Reduced Description of Physics

With the reduced operators $(M_0, M_1, \delta_0)$ defined above, we express our learned PDE directly as a reduced finite element system whose structure is inherited from the input mesh: $\delta_0$ encodes the induced (dense) control-volume topology, while $M_0$ and $M_1$ encode metric information through discrete inner products.

For a flux $\mathbf{J}$ and nonlinearity $\mathcal{F}_\theta$ and parameter $\epsilon > 0$, we discretize the following conservation law ansatz

$$\nabla \cdot \mathbf{J} = f \tag{6}$$

$$\mathbf{J} = \epsilon \nabla u + \mathcal{F}_\theta. \tag{7}$$

Following a standard mixed Galerkin discretization, we obtain the discrete flux network

$$M_1 J_\theta(u, z) = \varepsilon \delta_0 u + M_1 \mathcal{F}_\theta(u, z), \tag{8}$$

where $\mathcal{F}_\theta$ is a conditional transformer mapping between degrees of freedom conditioned on geometry. This yields the discrete system:

$$\underbrace{\varepsilon \delta_0^\top M_1 \delta_0 u + \delta_0^\top \mathcal{F}_\theta(u, z) - M_0 f_\theta(z)}_{G_\theta(u,z)} = 0. \tag{9}$$

Equivalently, defining the reduced Laplacian $K := \delta_0^\top M_1 \delta_0$, (9) corresponds to diffusion with a geometry-conditioned nonlinear flux term. The diffusive component $\varepsilon K u$ plays the role of an artificial viscosity that provides a stable linear backbone, while learning is restricted to the nonlinear flux contribution $\mathcal{F}_\theta$. Our implementation is described in Section 3.3.

### 2.4 PDE-Constrained Learning Problem

Given an $\ell_2$ reconstruction loss $\mathcal{L}(u) = \sum_d |u(x_d) - u_{data}(x_d)|^2$ we pose the equality constrained optimization problem:

$$\min_{u,\theta,\lambda} \mathcal{L}(u) + \lambda \cdot G_\theta(u, \alpha), \tag{10}$$

$$\text{s.t. } G_\theta(u, \alpha) = 0 \tag{11}$$

At each epoch, the Karush-Kuhn-Tucker conditions prescribe a nonlinear forward solve for $u$ and a linear adjoint

solve for $\lambda$. To update the model parameters $\theta$, gradients only need to be propagated through the final Newton iteration for $u$ via implicit differentiation (see Section 3.3).

## 3 General-Geometry Neural Whitney Forms (Geo-NeW)

With the learning problem defined, we present the primary contributions of this work: **1.** the encoding of mesh geometry so that the physical model outlined in (10) can generalize across unseen geometries, **2.** design of transformers to preserve required properties of the finite element construction, as well as **3.** stability analysis and an associated training algorithm which ensures the equality constraints in (10) have a well-posed solution throughout training. This approach builds on the CNWF formulation of Kinch et al. (2025) as the structure-preserving operator backbone and extends this work with a geometry-dependent basis and stable parameterization. The method is described in Figure 3.

To link the geometry information to the finite element basis and physics description, we will prescribe an encoding of geometric features describing $g$ to the conditioning variable $z$. In what follows, we assume $\mathcal{M}_g$ to be a shape regular, simplicial mesh of $g$ with a disjoint partition of its boundary into sidesets $\partial\Omega_g = \bigcup_{i=1}^{N_\Gamma} \Gamma_g^{(i)}$.

**Mesh Input.** We assume the following precomputed features that may be easily calculated for each $\mathcal{M}_g$ as a preprocessing step.

- **Patch labels.** For each side set of the boundary, $\Gamma_g^i$ is labeled with a one-hot feature specifying the type of boundary (e.g., "wall", "inlet", "outlet").
- **FEM matrices.** We construct both metric matrices $(M_0, M_1)$ and topological matrices ($\delta$) from the mesh, this is described in detail in (Kinch et al., 2025) and Appendix B.
- **Heat kernel signature (HKS).** The HKS (Sun et al., 2009) is an intrinsic spectral descriptor derived from the mesh Laplacian, defined per node by evaluating the heat kernel at multiple diffusion times. The HKS provides a multi-scale representation of local-to-global geometric context that is *invariant to rigid motion* and robust to discretization.
- **Harmonic coordinates (HC).** HC offers a minimal-complexity implementation of intrinsic harmonic cage coordinates (Joshi et al., 2007), constructed directly from the physically relevant boundary partitions. Cage coordinates provide a coordinate-free means of representing the geometry and encoding distances between interior partitions and boundaries through the graph Laplacian.
- **Signed distance function (SDF).** The SDF provides a measure of closeness to boundaries useful for describing boundary layers.

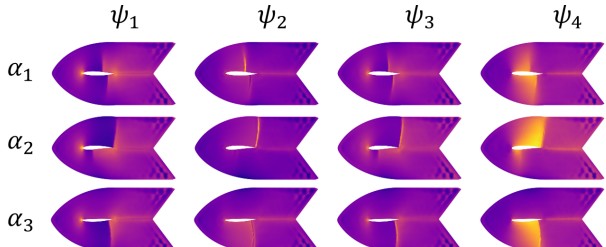

*Figure 3.* We demonstrate the adaptability of the learned finite element basis functions to variable geometries for airfoils with shocks. The shape functions track the location of the geometry-dependent shock, allowing accurate representations even at very low reduced dimension.

Together, these features provide a translationally and rotationally-invariant representation of $g$ that describes physics and geometry in a coordinate-free way, with several features (labels, HKS, HC, SDF) defined consistently across discretizations $\mathcal{M}_g$ of the domain $g$. The coordinate-free description of physics is a hallmark of the finite element exterior calculus, accordingly, our approach constructs coordinate-free descriptors of both geometry and physics. We present ablation results on these encodings in Section 4 and further description in Appendix C.

**Geometry encoder.** The geometry encoder $E_\theta$ maps the input mesh derived features $z_{mesh} = \{HKS, HC, SDF\}$ to a per-node context embedding that incorporates both local geometry and global shape information

$$E_\theta(z_{mesh}) = z \in \mathbb{R}^{N \times d_{model}},$$

where $d_{model}$ is the embedding dimension. This context conditions both (i) the geometry-conditioned, reduced FE spaces $\{\mathcal{W}_g^i(z)\}_i$ and (ii) the nonlinear PDE operator $\mathcal{P}_\theta(u, z)$. This encoder is modular: any mesh-to-vector model that produces node embeddings (graph neural network, transformer on mesh tokens, etc.) can be used.

While Geo-NeW admits any mesh-to-token geometry encoder, we require: (i) sub-quadratic scaling $O < (N^2)$, (ii) permutation equivariance over mesh tokens, (iii) support for variable input size $N$, and (iv) sufficient capacity to capture global geometric context. We implement an inducing-point *anchor transformer* encoder (Jaegle et al., 2021; Lee & Oh, 2024; Lee et al., 2019; Alkin et al., 2024). In each attention block, we sample $M \ll N$ *anchor tokens* from the $N$ mesh tokens, apply residual self-attention among anchors, and then update all mesh tokens via residual cross-attention from tokens to anchors. This yields per-node embeddings that incorporate both local detail and global shape information with $O(NM + M^2)$ memory and compute (Appendix E.2).

### 3.1 Geometry Conditioning

The encoded state $z$ provides per-node feature embeddings capturing both local geometry and global shape. We use $z$ to condition the partition $W_\theta$, flux $\mathcal{F}_\theta$, and source $f_\theta$

models via cross-attention pooling. For each sub-model, we perform Perceiver-style pooling using a learned set of latent query tokens $L_{\{W,\mathcal{F},f\}} \in \mathbb{R}^{n_c \times d_{\text{model}}}$, producing $n_c$ *context* tokens

$$c_{\{W,\mathcal{F},f\}} = \text{Attn}\big(L_{\{W,\mathcal{F},f\}}, z, z\big) \in \mathbb{R}^{n_c \times d_{\text{model}}},$$

which provides a compact geometry-conditioned summary used to parameterize the corresponding components.

## 3.2 Learned Partition-of-Unity Basis

The transformer $W(x, z)$ prescribing the reduced basis $\mathcal{W}_g^0$ is the core geometric object in our model, defining the control-volume discretization on which the learned PDE is solved. The representation is discretization invariant in that the field is specialized to a given mesh by evaluating $W(x_i, z)$ on the nodes of $\mathcal{M}_g$. We highlight that this decouples the *geometric conditioning* from the *basis evaluation*– different discretizations of the same mesh could be used.

Given per-node embeddings $z_{\text{mesh}} \in \mathbb{R}^{N \times d_{\text{model}}}$ and context tokens $c_W \in \mathbb{R}^{n_c \times d_{\text{model}}}$, we broadcast $c_W$ to each node, concatenate, and apply a weight-shared MLP with a linear skip connection to produce logits $S \in \mathbb{R}^{P \times N}$, a full description is provided in Appendix E.3. We then enforce the partition-of-unity constraint by applying a softmax across control volumes for each node, $W_{:,j} = \text{softmax}\big(S_{:,j}\big)$, which guarantees $W_{ij} \geq 0$ and $\sum_i W_{ij} = 1$. This can be viewed as a geometry-conditioned neural field on the query mesh nodes (Serrano et al., 2024; 2023; Qi et al., 2017). The reduced inner products are then computed from the fine operators as $M_0(z) = W_\theta(z) M_0'(z) W_\theta(z)^\top$ (and similarly for $M_1$, see Appendix B.3). For vector fields, we construct a per-field basis defined similarly, see Appendix E.3.

## 3.3 Stability, Uniqueness, and Implicit Differentiation

We train Geo-NeW via PDE-constrained optimization by solving the learned finite element system $G_\theta(u, z) = 0$ in the forward pass. For implicit differentiation to be well-defined and numerically stable, the nonlinear system must be locally well-posed. In particular, the operator should be coercive so that the solution $u^\star$ is locally unique and the Jacobian is invertible.

Following Kinch et al. (2025) Theorem 2.1, consider our flux parameterization (6) with reduced Laplacian $K$. If the nonlinear flux term $\mathcal{F}_\theta(u, z)$ is Lipschitz in $u$ with constant $C_L$ and $K$ is invertible (which holds after enforcing Dirichlet constraints, yielding an SPD operator), then the perturbed system admits a unique solution whenever

$$\tau = \varepsilon\, C_L \, \|K^{-1}\| < 1. \tag{12}$$

This motivates our flux parameterization, which explicitly enforces a uniform Lipschitz bound to preserve well-posedness across geometries (Section 3.3).

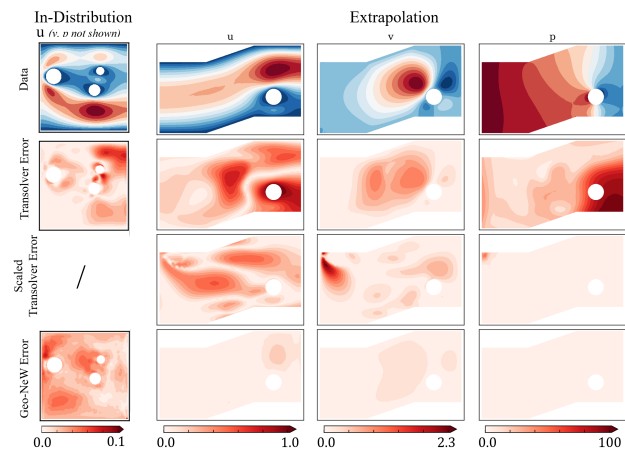

*Figure 4.* Geo-NeW's physical and geometric biases improve prediction of steady fluid flow. For models trained on a flow past obstacles in a square domain, we gently perturb the domain to demonstrate out-of-distribution geometry's effect on pointwise error for velocity $(u, v)$ and pressure $p$. **Row 1.** Target solutions. **Row 2.** Prediction by Transolver gives large concentrations of pointwise error. **Row 3.** Scaling the coordinate of positional embedding for Transolver to match unit cube of training data yields lower error but with hallucinated obstacles. **Row 4.** Geo-NeW predicts under extrapolation despite only having trained on a unit square.

Under (12), the Jacobian $J(u^\star) = \partial_u G_\theta(u^\star, z)$ is invertible, and $u^\star(\theta)$ is locally differentiable by the implicit function theorem. Gradients are computed via the adjoint method by solving $J(u^\star)^\top \lambda = \partial_{u^\star}\mathcal{L}$ and evaluating $\frac{d\mathcal{L}}{d\theta} = -\lambda^\top \partial_\theta G_\theta$ (Amos & Kolter, 2017; Agrawal et al., 2019).

**Lipschitz-bounded conditional flux model.** To satisfy the uniqueness condition in Section 3.3, we parameterize the learned nonlinear *dual* flux term $\mathcal{F}_\theta(u, z) \in \mathbb{R}^{P_1 \times F}$ with an explicit Lipschitz bound in $u$. We write it as a metric-weighted primal flux,

$$\mathcal{F}_\theta(u, z) := H_\theta(z)\, F_\theta(u, z), \tag{13}$$

where $F_\theta(u, z) \in \mathbb{R}^{P_1 \times F}$ is a primal edge-flux model and $H_\theta(z) \in \mathbb{R}^{P_1 \times P_1}$ is a geometry-conditioned SPD map playing the role of a reduced Hodge operator on edge quantities. We enforce $H_\theta(z) \succ 0$ via the parameterization $H_\theta(z) = \mathcal{H}(z)\, \mathcal{H}(z)^\top + \varepsilon_H I$, with $\varepsilon_H > 0$. For computational efficiency across many geometries, we learn $H_\theta(z)$ rather than using the explicit reduced $M_1$ in the original formulation (Kinch et al., 2025).

The primal flux $F_\theta$ is implemented as an antisymmetric edge function to preserve discrete conservation, i.e., $F_{ij}(u, z) = -F_{ji}(u, z)$. Geometry enters through a hypernetwork conditioned on the context tokens $c_F$, which produces the weights of an edge-wise MLP acting on reduced edge features (Ha et al., 2016; Chauhan et al., 2024). This design separates geometry conditioning (in $z$) from state dependence (in $u$), allowing expressive adaptation across geometries while enforcing hard Lipschitz control in $u$.

**Heat Transfer**      **Solid Mechanics**                 **Fluid Mechanics**

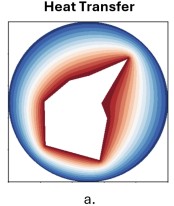 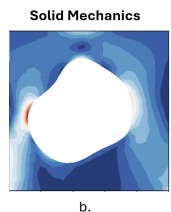 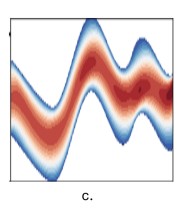 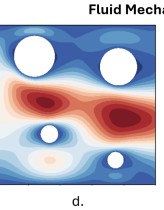 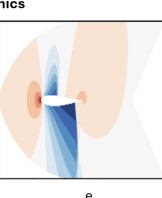 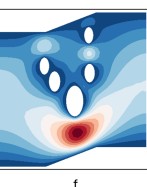

a.             b.               c.             d.            e.             f.

*Figure 5.* We consider six example problems consisting of canonical steady state PDEs across heat transfer(a.), solid mechanics (b.), and fluid mechanics (c.-f.). In four of these problems, Geo-NeW provides the best performance out of our baselines and those available from previous work.

We define the Lipschitz constant $C_L$ with respect to $u$ under $\ell_2$ norm by $\|\mathcal{F}_\theta(u_1, z) - \mathcal{F}_\theta(u_2, z)\| \leq C_L \|u_1 - u_2\|$, and enforce $C_L \leq \zeta$ with $\zeta < \left(\varepsilon\|K^{-1}\|\|\delta_0\|\right)^{-1}$, so that the uniqueness condition in (12) holds. In practice, we compute $\zeta$ from the reduced stiffness operator $K$ and discrete coboundary and update this bound during training (See Appendix E.4 for details of the matrix-norm estimate).

### 3.4 Implementation Details

We solve the resulting system with efficient batched Newton iterations through automatic differentiation. We sample initial states as $u_0 \sim \mathcal{N}(0,1)$. The solve is batched and GPU-compatible. Our framework optionally includes a non-conservative source or body force term, $f_\theta(z)$ (Shaffer et al., 2025). If a forcing is known, it can be included directly, or the learnable version may be used if it is not known, when unmodeled forcing may be present. For all evaluations herein, the source is excluded. Dirichlet boundary conditions are specified by identifying boundary node sets $\Gamma_D \subset \partial\Omega$. In all the learned finite element problems (9), boundary nodes are eliminated or constrained explicitly. This yields reduced operators acting on the interior subspace, ensuring coercivity and invertibility of the stiffness matrix, as well as exact satisfaction of Dirichlet BCs. Additional implementation details are described in Appendix E.

## 4 Results

We evaluate our models using normalized L2 error:

$$\epsilon = \frac{1}{N_{samples}} \sum_{i=1}^{N_{samples}} \frac{||u_i' - u_i||_2}{||u_i||_2}.$$

and train with the SOAP optimizer (Vyas et al., 2024). Details on training procedures are provided in Appendix E. Briefly, we use constant encoder hyperparameters across all evaluations with 4 encoder blocks and a model dimension of 128, resulting in Geo-NeW models with $\sim 3M$ parameters. We select the trainable reduced dimension from $P \in \{16, 32\}$ to balance computational complexity and expressivity, and use a maximum learning rate of $1e-3$ with cosine annealing. Where baseline results were not available, we followed the provided implementations and hyperparameter suggestions (Appendix I). We evaluate our Geo-NeW

method on both standard benchmarks for steady-state PDEs on variable geometries and on custom datasets designed to evaluate out-of-distribution evaluation. For baseline models, we consider point-cloud-based inputs consisting of nodal coordinates and optionally the SDF if specified in the original implementation. We evaluate on standard benchmarks provided in the Geo-FNO (Pipe, Elasticity, NACA) (Li et al., 2023b), and NS2d-c (Hao et al., 2023) as well as custom datasets to demonstrate out-of-distribution generalization.

### 4.1 Standard Benchmarks

We first consider three standard variable problems from Li et al. (2023b). We only consider those problems which are steady state and *variable geometry*, these are pipe flow, elasticity for a unit cell with a variable cavity, and Euler flow around a variable shape airfoil (see Appendix G). The NACA and Elasticity problems target a derived quantity; we impose appropriate boundary conditions to form a well posed effective model for consistent comparison. The results on these datasets are shown in Table 2. Notably, for elasticity and pipe flow, we outperform all previous ML-based methods that we are aware of, with an improvement of 29% and 71%, respectively. We also evaluate the training throughput in Appendix H.

### 4.2 Out-of-Distribution Generalization

We consider two custom evaluations to highlight the capability for inference on novel geometries. The **Poly-Poisson** dataset consists of circular domains with a polygon cutout. We solve Poisson with Dirichlet BCs on the cutout and exterior, and vary the degree of the polygon to create a distribution shift. Models are trained on $n < 5$ and evaluated on $n > 5$. Because this benchmark is governed by diffusion alone, this problem matches the hypothesis class of the Geo-NeW method by design, and we therefore expect good performance. Results are shown in Figure 4, our approach provides an improvement over conventional baselines both in and out-of distribution.

We also consider an extension of the 2D Navier-Stokes data from Hao et al. (2023), with some extensions to allow larger variation over geometries (**NS2d-c++**). For training data, we consider only geometries with circular obstacles, in a unit square, with $n < 4$ obstacles. For extrapolation, we evaluate

*Table 1.* Prediction error across **Non-standard** PDE datasets (lower is better).

| Model $\quad 1e-2\downarrow$ | Poly ID | Poly OOD | NS2d-c | NS2d-c++ ID | NS2d-c++ OOD | NS2d-c++ OOD (mild) |
|---|---|---|---|---|---|---|
| GNOT (2023) | 0.074 | 5.24 | 0.93 | 5.30 | 83.1 | 14.1 |
| Transolver (2024) | 0.064 | 7.04 | **0.63** | 4.04 | 91.4 | 15.1 |
| Linear Attention (2020) | 0.055 | 4.60 | 0.66 | 3.11 | 91.8 | 13.1 |
| Inducing Point (baseline) | 0.054 | 8.90 | 1.82 | 4.47 | – | – |
| **Ours (Geo-NeW)** | **0.033*** | **2.14*** | 1.10 | **1.93** | **42.2** | **7.87** |

\* does not use harmonic coordinates

*Table 2.* Prediction error across **Standard** PDE datasets (lower is better).

| Model $\quad 1e-2$ | NACA $\downarrow$ | Elasticity $\downarrow$ | Pipe $\downarrow$ |
|---|---|---|---|
| U-Net (2015) | 0.79 | 2.35 | 0.65 |
| DeepONet (2021) | 3.85 | 9.65 | 0.97 |
| Geo-FNO (2023) | 1.38 | 0.67 | 0.74 |
| Galerkin (2021) | 1.18 | 2.40 | 0.98 |
| GNOT (2023) | 0.76 | 0.86 | 0.47 |
| Transolver (2024) | 0.53 | 0.64 | 0.46 |
| LaMO (2025) | **0.41** | 0.50 | 0.38 |
| **Ours (Geo-NeW)** | 0.82† | **0.35†** | **0.11** |

† modeling derived quantity

*Table 3.* Dirichlet boundary error on NS2d-c in normalized L2.

| Method | $u$ | $v$ | $p$ |
|---|---|---|---|
| Baseline encoder | 2.22e-3 | 2.99e-3 | 7.58e-3 |
| **Geo-NeW (ours)** | **0.00** | **0.00** | **0.00** |

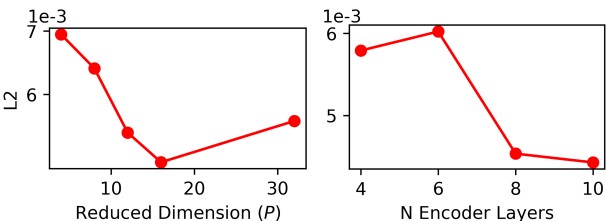

*Figure 7.* We evaluate the L2 error as a function of reduced dimension and encoder depth, highlighting the tradeoff between expressivity and computational cost.

**Mesh encoding ablation.** We consider both the baseline inducing point encoder and the Geo-NeW model with an identical encoder implementation on the point-cloud and mesh-based input encodings. These results are shown in Table 4, and highlight the value of structure-preserving inductive bias beyond the benefit of mesh-based data processing, which is also substantial.

*Table 4.* Prediction error (lower is better) comparing model family and input encoding on NS2d-c++ in-distribution test after 300 epochs for all models.

| | Point-cloud | Mesh-based |
|---|---|---|
| Baseline encoder | 8.82e-2 | 7.21e-2 |
| **Geo-NeW (ours)** | 6.76e-2 | **5.17**e-2 |

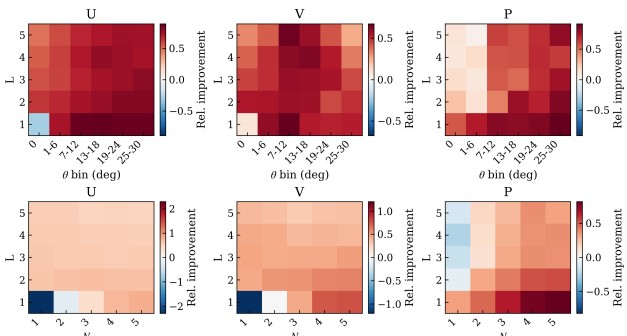

*Figure 6.* Geo-NeW provides a substantial improvement in geometry extrapolation over Transolver in our settings. We show the relative improvement between the methods over a range of geometry parameters ($L$, $N_{obs}$, $\theta$).

## 5 Conclusion

We introduced a geometry-conditioned, structure-preserving implicit neural PDE solver, that achieves competitive in-distribution accuracy and distinct advantages for out-of-distribution inference on unseen geometries. This results from explicit enforcement of conservation laws and boundary conditions by construction. While demonstrated on 2D steady-state problems, the framework extends naturally to 3D. We position this work as a step toward physics and geometry-aware foundation models that couple learned operators with domain knowledge. Geometry generalization is essential for design applications, where the geometries of interest are precisely those not available during training.

models with $n \leq 6$ obstacles, on geometries with variable length $L \leq 5$, with square obstacles, and with an angled step with angle $\theta \leq 30°$. Coordinate-based baseline models fail to produce stable predictions under these extrapolation settings, yielding normalized $\ell_2$ errors exceeding $\varepsilon > 30\%$. Restricting the evaluation to $L = 1$ while varying step angle and obstacle parameters yields a milder extrapolation regime in which Geo-NeW achieves $7.87\%$ error compared to $13.1\%$ for the strongest baseline. We verify the exact treatment of boundary error in Table 3.

Results are shown in Figures 6 and 4. Across all extrapolation variables, Geo-NeW consistently outperforms baselines, with the largest gains from increasing obstacle count and introducing the angled step. We provide full results in Table 1.

## Limitations

Our experiments are restricted to two-dimensional steady-state PDEs and assume access to a computational mesh and specified boundary conditions. Geo-NeW also incurs additional solve cost relative to direct regression models, and its stability constraints restrict the learned constitutive model class.

## Impact Statement

This work advances neural PDE surrogates for engineering design by improving generalization to unseen geometries while preserving physical structure. As with other learned simulators, predictions should be validated with trusted numerical methods before use in safety-critical design or decision-making.

## Acknowledgments

The authors acknowledge support from the Department of Energy, Office of Science, Advanced Scientific Computing Research (ASCR) program SEA-CROGS MMICCs center (DE-SC0023191), as well as Army Research Office support through the "Complexity, Nonlocality, and Uncertainty in Heterogeneous Solids" MURI program (W911NF-24-2-0184). B.S. is supported by the National Science Foundation (NSF) Graduate Research Fellowship (DGE-2236662). M.A.H. is supported by NSF Award 2121887.

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

# A Summary of Notation used

| Symbol | Description | Symbol | Description |
|---|---|---|---|
| $\Omega_g$ | Domain for geometry $g$ | $M_g$ | Mesh of $\Omega_g$ |
| $\partial\Omega_g$ | Boundary (with sidesets) | $z$ | Geometry-conditioned context |
| $u$ | PDE state (reduced coeffs) | $F$ | Number of fields |
| $\phi_j$ | Fine nodal basis | $\phi^1_{ab}$ | Fine Whitney–1 basis |
| $W$ | Nodal projection ($P \times N$) | $W_1$ | Edge projection ($P_1 \times N_1$) |
| $\psi^0_i$ | Reduced nodal basis | $\psi^1_{ij}$ | Reduced Whitney–1 basis |
| $M'_0,\ M'_1$ | Fine mass matrices | $M_0,\ M_1$ | Reduced mass matrices |
| $\delta'_0,\ \delta_0$ | Fine / reduced incidence | $K$ | Reduced stiffness |
| $\mathcal{F}_\theta$ | Learned flux | $C_L$ | Lipschitz constant |
| $\varepsilon$ | Diffusion parameter | $\zeta$ | Lipschitz bound |
| $G_\theta(u,z)$ | PDE residual | $\mathcal{L}(u)$ | Training loss |
| $\lambda$ | Adjoint variable | | |

# B Additional mathematical background

Adapted from (Kinch et al., 2025), we present some mathematical background on FEEC and learnable Whitney forms to provide additional context for this work. Interested readers should refer to previous work (Trask et al., 2022; Kinch et al., 2025; Actor et al., 2024; Arnold, 2018).

We briefly summarize the discrete differential-operator structure used in this work. We consider a bounded Lipschitz domain $\Omega \subset \mathbb{R}^d$ with boundary $\partial\Omega$ and outward unit normal $\hat{n}$.

## B.1 Whitney finite element spaces

Let $\{\lambda_i\}_{i=1}^N$ denote a *partition of unity* on $\Omega$, satisfying $\lambda_i \geq 0$ and $\sum_{i=1}^N \lambda_i = 1$. In standard low-order FEEC on simplicial meshes, $\lambda_i$ are the nodal $\mathcal{P}_1$ barycentric basis functions.

The associated lowest-order Whitney spaces can be written in terms of $\{\lambda_i\}$ as

$$\mathcal{W}_0 = \mathrm{span}\{\lambda_i\}, \tag{14a}$$

$$\mathcal{W}_1 = \mathrm{span}\{\lambda_i \nabla\lambda_j - \lambda_j \nabla\lambda_i\}, \tag{14b}$$

$$\mathcal{W}_2 = \mathrm{span}\{\lambda_i \nabla\lambda_j \times \nabla\lambda_k + \lambda_j \nabla\lambda_k \times \nabla\lambda_i + \lambda_k \nabla\lambda_i \times \nabla\lambda_j\}. \tag{14c}$$

(higher-degree forms $\mathcal{W}_2, \mathcal{W}_3$ can be defined analogously.) These spaces may be identified with the standard compatible finite element sequence: $\mathcal{W}_0$ corresponds to nodal $P_1$ elements, $\mathcal{W}_1$ corresponds to Nédélec edge elements, and $\mathcal{W}_2$ corresponds to Raviart-Thomas face elements.

## B.2 Discrete de Rham complex and exactness

Given any barycentric coordinate $\lambda$ (e.g. either the fine scale basis $\phi$ or the reduced basis $\psi^0$ defined in the paper), the construction above prescribes the following de Rham sequence:

$$
\begin{array}{ccccccc}
H(\mathrm{grad}) & \xrightarrow{\ \nabla\ } & H(\mathrm{curl}) & \xrightarrow{\ \nabla\times\ } & H(\mathrm{div}) & \xrightarrow{\ \nabla\cdot\ } & L^2 \\
\big\uparrow & & \big\uparrow & & \big\uparrow & & \big\uparrow \\
\mathcal{W}_0 & \xrightarrow{\ d_0\ } & \mathcal{W}_1 & \xrightarrow{\ d_1\ } & \mathcal{W}_2 & \xrightarrow{\ d_2\ } & \mathcal{W}_3
\end{array}
\tag{15}
$$

The linear maps $d_k$ are the *discrete exterior derivatives* and serve as mesh-compatible analogues of gradient, curl, and divergence. They satisfy the exactness property

$$d_{k+1}\, d_k = 0, \tag{16}$$

which mirrors the vector calculus identities $\nabla \times \nabla = 0$ and $\nabla \cdot (\nabla\times) = 0$. This compatibility is the algebraic mechanism underlying discrete conservation properties.

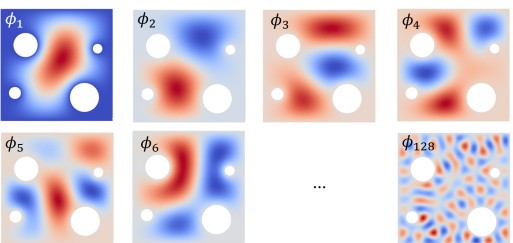

*Figure 8.* Eigenvectors of the graph Laplacian for NS2d-c sample mesh.

### B.3 Local antisymmetric fluxes and discrete conservation

A key structural feature for conservation laws is that Whitney 1-forms induce *pairwise antisymmetric flux exchanges* between partitions. To see this, consider a flux field $J \in \mathcal{W}_1$ and test with $q = \lambda_i \in \mathcal{W}_0$. Using only the partition-of-unity identities

$$\sum_i \lambda_i = 1, \qquad \sum_i \nabla \lambda_i = 0, \tag{17}$$

the term $(J, \nabla \lambda_i)$ may be expanded as

$$(J, \nabla \lambda_i) = \sum_j \int_\Omega (\lambda_i \nabla \lambda_j - \lambda_j \nabla \lambda_i) \cdot J \, dx. \tag{18}$$

The integrand is antisymmetric under $i \leftrightarrow j$, i.e.,

$$\lambda_i \nabla \lambda_j - \lambda_j \nabla \lambda_i = -(\lambda_j \nabla \lambda_i - \lambda_i \nabla \lambda_j).$$

Consequently, when summing (18) over all $i$, the internal flux contributions cancel in pairs, leaving only boundary flux terms. This cancellation is the discrete analogue of local conservation in divergence form.

**Projection of discrete operators.** Let $\{\phi_j\}_{j=1}^N$ denote the fine nodal basis and $\{\phi_{ab}^1\}_{(a,b)\in E}$ the associated Whitney–1 basis $\phi_{ab}^1 = \phi_a \nabla \phi_b - \phi_b \nabla \phi_a$. The learned reduced nodal basis is defined by

$$\psi_i^0(x) = \sum_{j=1}^N W_i(x_j) \, \phi_j(x), \qquad W \in \mathbb{R}^{P \times N}.$$

The reduced 0-form mass matrix is obtained by,

$$M_0 = W \, M_0' \, W^\top,$$

where $M_0'$ is the fine nodal mass matrix. The associated reduced Whitney-1 basis ((14) induces an edge projection $W_1 \in \mathbb{R}^{P_1 \times N_1}$ with entries

$$(W_1)_{(ij),(ab)} = W_{ia} W_{jb} - W_{ib} W_{ja}.$$

in matrix form. Let $\delta_0'$ and $\delta_0$ denote the fine and reduced node–edge incidence matrices, respectively. The induced projection satisfies

$$\delta_0' W^\top = W_1^\top \delta_0.$$

The reduced 1-form mass matrix is then

$$M_1 = W_1 \, M_1' \, W_1^\top,$$

where $M_1'$ is the fine Whitney–1 mass matrix.

## C Mesh Derived Feature Encoding

The graph Laplacian provides a rich and efficient means for intrinsic geometry processing (Crane et al., 2013; Reuter et al., 2006)

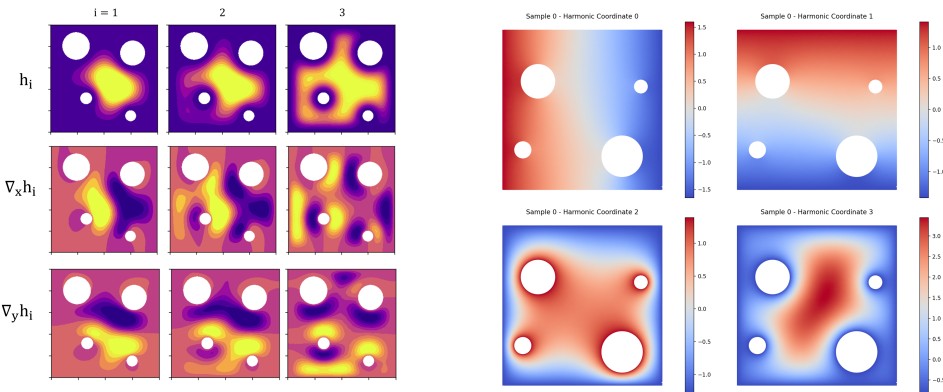

*Figure 9.* We showcase the Heat Kernel Signature and spatial gradients (left) and Harmonic Coordinates (right) for a given mesh.

## C.1  Spectral features via heat kernel signature

The mesh enables the solution of intrinsic spectral problems. On the interior (non-boundary) subspace, we solve the generalized eigenproblem $K_{interior}\,\phi_k = \lambda_k\,M_{interior}\,\phi_k$. Using the Laplace eigenpairs, we compute the heat kernel signature at each node:

$$\text{HKS}(x,t) = \sum_{k=1}^{K} \phi_k(x)^2\,e^{-t\lambda_k}, \tag{19}$$

for a set of diffusion times $\{t_j\}$. The HKS provides a multiscale intrinsic descriptor that is invariant under isometries and robust to mesh refinement. These features serve as geometry-aware tokens for the encoder.

We additionally include the spatial gradients of the spectral HKS features, $\nabla\phi$, which encode direction geometric information and can be trivially computed on the meshed representation. Within the distribution of geometries considered, HKS provides a stable multiscale intrinsic signature that is empirically discriminative.

In general, no purely spectral descriptor can be guaranteed injective over all planar domains due to isospectral counterexamples; however, such collisions are nongeneric, and our empirical evaluation shows separation over the considered geometry family (Datchev & Hezari, 2011).

## C.2  Harmonic coordinates from boundary groups

When boundary components are labeled, the mesh allows the construction of harmonic coordinate functions. For a disjoint boundary partition of the form $\partial\Omega = \bigcup_{i=1}^{N_{groups}} \Gamma_i$ we construct harmonic coordinates from pairs of boundary groups $(\Gamma_i, \Gamma_j)$. We solve

$$\Delta\psi = 0 \quad \text{in } \Omega, \qquad \psi = 1 \text{ on } \Gamma_i, \qquad \psi = 0 \text{ on } \Gamma_j. \tag{20}$$

Each solution $\psi$ provides a smooth coordinate aligned with the geometry and boundary configuration. Multiple such solves yield a low-dimensional harmonic embedding of the domain that is sensitive to topology and boundary layout.

These functions can be interpreted as boundary-conditioned harmonic extensions induced by the discrete Laplace–Beltrami operator, closely related to Laplacian-based embeddings and intrinsic coordinate constructions. Unlike spectral coordinates, they are localized, boundary-aware, and invariant to rigid transformations, making them well suited for geometry-general PDE learning. Since assembly of the discrete Laplacian ($K$) is already required by our method, these coordinates incur negligible additional cost. Unlike generic graph or point-cloud settings, scientific computing pipelines necessarily provide labeled boundary partitions as part of PDE specification, making boundary-conditioned constructions a natural and broadly applicable design choice in SciML (Zienkiewicz et al., 1977).

To demonstrate the intrinsic nature of the model under the harmonic coordinate definition, we consider a zero-shot generalization to scenarios where the NS2d-c data flow direction is reversed. Using the identication of one boudnary partition with the inlet and one with the outlet, we can evaluate the model for flow in the reverse direction by swapping this assignment. As shown in Figure 10, we automatically obtain a qualitatively reasonable prediction with pressure gradients and vertical velocities corresponding to the flow in the reversed direction.

## D  Additional Results

While a Lipschitz limit on the flux network does provide guaranteed theoretical well-posedness of the learned PDE constraint, this does not guarantee that this solution can be found, or found quickly, from a given initialization under Newton's method.

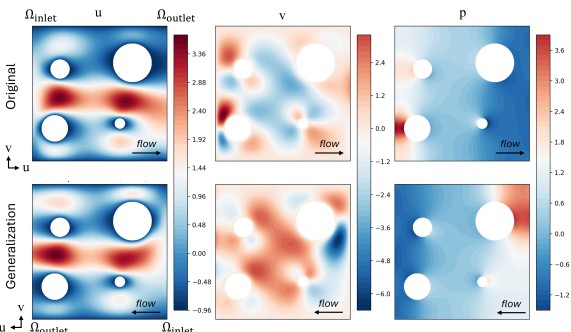

*Figure 10.* Swapping of the input and output labels without providing any training data provides a semantically meaningful prediction of flow from inlet to outlet.

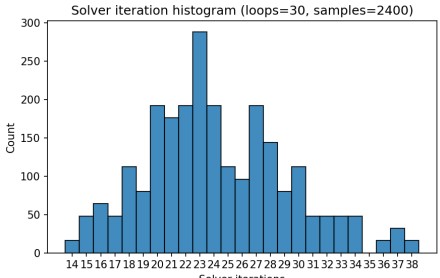 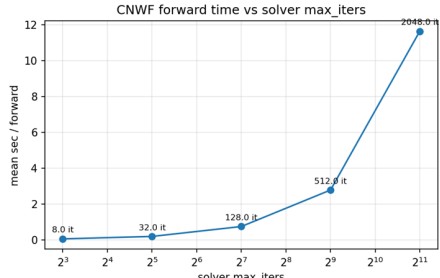

*Figure 11.* Solving the reduced finite element problem in few iterations is essential to maintain throughput (right), in practice we maintain stable solves within modest iterations, for instance in the elasticity problem (left).

Training requires a full solve of the reduced system at each iteration, accordingly the computational cost of training scales greatly with the needed iterations for convergence. A key challenge is therefore to pose the learned system in a manner that not only provides a unique solution, but does so in minimal iterations (for instance, less than 10).

**Stability** We batch the Newton solve operations on the GPU. This results in different fluxes, $F(\hat{u}_i, z_i)$ for each sample $i$ in the batch. To prevent a particularly stiff system from resulting in higher solve iterations for all samples, we set a threshold for a percentage of samples in a given batch that have reached the solver tolerance. When this threshold has been reached, the converged samples are used to compute the loss and backpropagate. We set this threshold at $0.8$ for all experiments, and the maximum solve iterations at $200$ for a tolerance of $1e-6$ in single precision. In practice, we maintain convergence $> 99\%$ for most of the training across experiments.

# E  Model implementation details

## E.1  Implementation Details

Fine-mesh operators are assembled and stored in sparse format, while all reduced operators are dense due to the overlapping control-volume construction; projections between these representations are implemented using sparse–dense matrix multiplications for efficiency. Fine-mesh quantities are padded to enable batching across geometries with varying numbers of nodes, whereas the reduced representation has a fixed dimension $P$ and therefore requires no padding. The reduced discrete derivative operator $\delta_0$ depends only on the reduced space dimension and orientation conventions and is precomputed once and reused across all forward passes. Dirichlet boundary conditions are enforced by augmenting the partition-of-unity map with fixed boundary partitions provided as input, ensuring exact boundary constraints via an affine combination. The resulting reduced nonlinear system is solved using batched Newton iterations implemented with automatic differentiation in PyTorch, enabling end-to-end training via implicit differentiation. Initial guesses for the nonlinear solve are sampled randomly; in rare cases of non-convergence, solves can be retried with different initializations. All experiments are conducted in single precision for improved throughput, though double precision may be preferable in applications requiring stricter conservation enforcement or higher solver accuracy, as solver tolerances impose a practical limit on achievable error.

## E.2  Inducing Point Geometry Encoder

For this work, we implement a minimal complexity anchor-based (or supernode) transformer using inducing point methods such as (Jaegle et al., 2021; Lee & Oh, 2024; Lee et al., 2019; Alkin et al., 2024). Empirically, this method provides

rich features and efficient scaling on both compute and memory usage. These approaches do not seek to approximate full self-attention and retain significant advantages in memory usage for large-scale problems. Let $\{z^{(i)}_{mesh}\}^{N}_{i=1}$, with $z^i_{mesh} \in \mathbb{R}^{d_{model}}$, denote feature embeddings associated with the $N$ nodes of a mesh coming from the mesh-based processing. Specifically, we select a small set of $M \ll N$ *anchor tokens* $\{a_j\}^{M}_{j=1}$ by randomly sampling node embeddings. The anchors first exchange information via self-attention,

$$\tilde{a} = a + \text{Attn}(a, a, a),$$

enabling global aggregation at cost $\mathcal{O}(M^2)$. The updated anchors then broadcast global context back to all nodes via cross-attention,

$$\tilde{z} = z + \text{Attn}(z, \tilde{a}, \tilde{a}),$$

yielding node features that combine local information with a low-rank global summary at cost $\mathcal{O}(NM)$. We stack multiple of these encoder blocks and resample the anchor points independently in each. Each attention block is followed by a position-wise feed-forward network, and all components are implemented in a pre-layer-normalized residual form. This design preserves per-node representations while providing scalable global information exchange, and empirically was well suited for our downstream tasks.

### E.3   W model

To improve optimization stability and help prevent degenerate partitions during early training, we parameterize the partition logits using a linear skip connection. Given node-wise inputs $z_{\text{in}} \in \mathbb{R}^{d_{\text{in}}}$ (formed by concatenating per-node geometry features and broadcast context), we compute control-volume logits as

$$S = W_{\text{mlp}}(z_{\text{in}}) \; + \; \alpha \, W_{\text{lin}}(z_{\text{in}}),$$

where $W_{\text{mlp}}$ is a multi-layer perceptron, $W_{\text{lin}}$ is a single linear layer, and $\alpha \in [0, 1]$ controls the relative contribution of the linear term. At initialization, we set $\alpha$ close to 1.

**Per-field basis.**   For systems with $F > 1$ (vector-valued) fields, $u = (u^{(1)}, \ldots, u^{(F)})$, we define the reduced spaces independently for each component. For each field $k \in \{1, \ldots, F\}$ we introduce a geometry-conditioned partition

$$\psi^{0,(k)}_i(x) \; = \; \sum^{N}_{j=1} W^{(k)}_i(x_j, z) \, \phi_j(x), \qquad \sum_i W^{(k)}_i(x_j, z) = 1,$$

yielding the reduced nodal space $\mathcal{W}^{0,(k)}_g = \text{span}\{\psi^{0,(k)}_i\}$. The associated edge space is defined as

$$\mathcal{W}^{1,(k)}_g = \text{span}\{\psi^{0,(k)}_i \nabla \psi^{0,(k)}_j - \psi^{0,(k)}_j \nabla \psi^{0,(k)}_i\}.$$

All reduced operators are block-diagonal across fields, while physical coupling is handled exclusively through the learned flux term.

### E.4   Flux model

We define the Lipschitz constant $C_L$ as the constant that bounds $||\mathcal{F}_\theta(u_1, z) - \mathcal{F}_\theta(u_2, z)|| \leq C_L ||u_1 - u_2||$. We prescribe a bound for $C_L$ as $\zeta < \tau(\varepsilon ||K^{-1}||)^{-1}$ to satisfy our uniqueness condition and aim to construct a model that guarantees $C_L < \zeta$. The value of $\zeta$ is computed directly from the reduced stiffness matrix and is updated periodically during training to maintain an accurate, but not universally guaranteed, bound.

The nonlinear flux $\mathcal{F}_\theta$ is implemented as a geometry-conditioned hypernetwork acting on reduced edge features (Ha et al., 2016; Chauhan et al., 2024). Let $(i, j)$ index an oriented edge in the reduced graph. For each edge, we form a projected feature on the mean and difference

$$\xi_{ij} = \left[ \Pi^\top (\hat{u}_i - \hat{u}_j), \; (-1)^{s(i,j)} \tfrac{1}{2} \Pi^\top (\hat{u}_i + \hat{u}_j) \right], \tag{21}$$

where $\Pi$ is a learned linear projection from field space to a lower-dimensional operator space, and $s(i, j) = 1_{i>j}$ is a sign operator to make the mean feature an odd-valued input.

Given a geometry context vector $c_F(z)$ produced by the encoder, the hypernetwork outputs operator matrices $A(c_F), B(c_F), C(c_F)$. These are computed from a geometry independent base and conditioned hypernetwork outwork as $A(c_F) = A_0 + A_{hyper}(c_F)$. Together with learned scalar amplitudes $\beta, \gamma$, the per-edge flux is defined with as,

$$F'_{ij} = \beta' A(c_F) \xi_{ij} + \gamma' C(c_F) \tanh\big(B(c_F) \xi_{ij}\big). \tag{22}$$

The hypernet operators ($A(c_F)$, $B(c_F)$, $C(c_F)$), projection operator $\Pi$, and SPD metric matrix $H(c_F)$ are spectrally normalized to give an upper-bounded Lipschitz constant of 1 at each evaluation. Additionally, the *tanh* function has a Lipschitz bound of 1. Therefore, this results in a system with a total Lipschitz constant of $\beta' + \gamma'$, which are derived from the learnable parameters as $\beta' = \min(\zeta/2, \beta)$, $\gamma' = \min(\zeta/2, \gamma)$, and therefore the maximum Lipschitz constant of the flux model as a whole is exactly $\zeta$. Since $\gamma$ controls the nonlinear contribution in particular, it plays a large role in the stability of the system as a whole. We compute $\gamma$ from the eigenvalue decomposition of the reduced stiffness matrix for a random subset of samples, and track a moving average during training. We apply a tolerance so the maximum Lipschitz constant is at most half of this value, which we find empirically improves stability and reduces the number of solve iterations. While this implementation does not guarantee the condition in (12) is satisfied for every sample, we find empirically that we maintain stable models in both training and validation. If a stronger stability guarantee is needed, $\gamma$ could be updated for every sample, at additional cost. We initialize all base operators to the identity and zero the last layer of the hypernetworks.

To bound the Lipschitz constant of the hypernetwork outputs, we upper bound the induced $\ell_2$ operator norm via the $\ell_\infty$ matrix norm. For each linear operator $A$, we compute

$$\|A\|_\infty = \max_i \sum_j |A_{ij}|,$$

which efficiently bounds the spectral norm as $\|A\|_2 \le \sqrt{n}\,\|A\|_\infty$ for $W \in \mathbb{R}^{n \times n}$ (Liu et al., 2022) while noting that this not usually a tight bound (Virmaux & Scaman, 2018) which may limit model expressivity .

The flux is constructed to be antisymmetric across edge orientation so that

$$F'_{ij}(\hat{u}, z) = - F'_{ji}(\hat{u}, z), \tag{23}$$

for every unordered edge $\{i, j\}$. This is achieved via an odd input embedding of the state, and maintaining odd functions via linear layers without biases.

### E.5 Dirichlet boundary conditions.

For each boundary group $\Gamma \subset \partial \Omega_g$ with prescribed Dirichlet data $u_b$, we augment the learned partition-of-unity with a boundary-adapted basis constructed directly from $u_b$. Let

$$s_\Gamma := \sum_{x_i \in \Gamma} u_b(x_i), \qquad \psi_\Gamma(x_i) := \frac{u_b(x_i)}{s_\Gamma}, \quad x_i \in \Gamma,$$

which defines a nonnegative, normalized boundary partition encoding the shape of the prescribed boundary values. To preserve the partition-of-unity on the boundary, we introduce the complementary partition

$$\psi_{\Gamma,\text{rest}}(x_i) := 1 - \psi_\Gamma(x_i), \quad x_i \in \Gamma.$$

The boundary values are then represented exactly as

$$u(x_i)\big|_\Gamma = s_\Gamma\, \psi_\Gamma(x_i),$$

while interior partitions remain unconstrained. In the reduced system, the coefficients associated with $\psi_\Gamma$ are fixed to $s_\Gamma$, and the corresponding rows and columns of the discrete operator are eliminated, enforcing Dirichlet conditions exactly while solving only for free interior degrees of freedom.

### E.6 Interpretation as physical graph-based network

From a graph-learning perspective, our method can be viewed as a conservative message passing network on a geometry-conditioned reduced graph. Nodes correspond to coarse partitions (Whitney 0-forms), and edges carry flux variables (Whitney 1-forms), so that discrete divergence corresponds to aggregating pairwise antisymmetric flux exchanges across neighboring partitions. Unlike standard graph neural surrogates that learn an explicit propagation rule through stacked message passing layers, Geo-NeW performs propagation implicitly by solving a globally coupled discrete balance law, while learning only constitutive components (nonlinear fluxes and sources) conditioned on geometry.

## F  Experiment details

All models are trained using SOAP (Vyas et al., 2024), with a cosine annealing learning rate scheduler. All models are trained and evaluated on a single Nvidia H200 GPU. Runtimes are reported based on this hardware.

| Dataset | Enc. layers | Enc. heads | Model dim | $P$ | Epochs | Max LR | Batch | $\gamma$ init (Flux) |
|---------|-------------|------------|-----------|-----|--------|--------|-------|-----------------------|
| NACA | | | | 32 | 1000 | | 16 | -2 |
| Elasticity | | | | 32 | 10k | | 16 | -4 |
| Pipe flow | 4 | 2 | 128 | 32 | 5k | 1e-3 | 16 | -2 |
| Poly-Poisson | | | | 8 | 2k | | 16 | -2 |
| NS2d-c | | | | 32 | 12k | | 16 | -2 |
| NS2d-c++ | | | | 16 | 1k | | 8 | -3 |

*Table 5.* Geo-NeW hyperparameters for all experiments. Shared parameters are merged for clarity.

| Dataset | $N$ (max mesh size) | Train samples | Test samples | OOD Samples |
|---------|---------------------|---------------|--------------|-------------|
| NACA | 11,271 | 1,000 | 100 | |
| Elasticity | 972 | 1,000 | 200 | |
| Pipe flow | 16,641 | 1,000 | 200 | |
| Poly-Poisson | 952 | 4,498 | 500 | 5,002 |
| NS2d-c | 12,000 | 1200 | 200 | |
| NS2d-c++ | 3,565 | 3,490 | 500 | 2478 |

*Table 6.* Dataset sizes and mesh resolutions for all Geo-NeW experiments.

### F.1 Hyper-parameters

Hyperparameters for the Geo-NeW models are reported in Table 5.

### F.2 Solver Configuration

The learned nonlinear system is solved using an iterative Newton method through pytorch automatic differentiation, with a maximum of 200 iterations and a convergence tolerance of 1e-6. Initial guesses are randomly sampled, and convergence is assessed independently for each element in a batch. Batches are accepted if at least 80% of solves converge within the iteration limit during training; non-converged instances may be retried with new initializations, and higher maximum iterations.

## G   Data details

Computational meshes are not available for every dataset. When needed, we use Delaunay triangulation on the provided interior point-cloud. Boundary groups are specified based on segments of the boundary condition definition and domain extent, automatically and manually verified. We use boundary conditions as described in the original data setup. The NACA and Elasticity benchmarks evaluate derived fields (Mach number and von Mises stress) rather than primary PDE states. For these tasks, Geo-NeW models the reported field directly as an effective surrogate and imposes Dirichlet conditions on the inlet and clamped boundary to define a well-posed effective boundary value problem. We use scikit-fem (Gustafsson & Mcbain, 2020) for internal finite element assembly (i.e. for mass matrices).

### G.1   NACA Airfoil

This benchmark considers steady two-dimensional compressible Euler flow over airfoils parameterized by the NACA-0012 family (Li et al., 2023a). Geometry variation is introduced through changes in the airfoil shape, while boundary conditions correspond to transonic inflow. The learning task is to predict the steady-state Mach number field on the mesh for each geometry.

### G.2   Pipe Flow

This dataset models steady incompressible Navier–Stokes flow,

$$-\nu\Delta\mathbf{u} + (\mathbf{u} \cdot \nabla)\mathbf{u} + \nabla p = \mathbf{0}, \qquad \nabla \cdot \mathbf{u} = \mathbf{0},$$

in two-dimensional pipe-like domains with smoothly varying centerline geometry (Li et al., 2023a). Geometry variation is controlled through polynomial perturbations of the pipe centerline. The output is the steady horizontal velocity field.

### G.3   Linear Elasticity on Unit Cell

This benchmark solves the steady linear elasticity equations,

$$-\nabla \cdot \sigma(\mathbf{u}) = 0,$$

on a unit square with a variable internal cavity (Li et al., 2023a). Geometry variation arises from changes in the cavity shape. The target quantity is the derived Von-Mises stress.

### G.4 Poly-Poisson

The Poly-Poisson dataset is a custom benchmark designed to probe geometric out-of-distribution generalization. We solve the steady Poisson equation,

$$-\Delta u = f,$$

on circular domains with an internal polygonal cutout. Dirichlet boundary conditions are imposed on both the exterior boundary and the polygon. Geometry variation is controlled by the number of polygon sides $n$. Models are trained on geometries with $n < 5$ and evaluated on geometries with $n > 5$, inducing a controlled distribution shift in geometric complexity.

### G.5 2D Steady-State Navier–Stokes (NS2d-c)

This benchmark is adapted from Hao et al. (2023) and considers steady incompressible Navier–Stokes flow,

$$-\nu\Delta\mathbf{u} + (\mathbf{u} \cdot \nabla)\mathbf{u} + \nabla p = \mathbf{0}, \qquad \nabla \cdot \mathbf{u} = 0,$$

in two-dimensional domains containing multiple circular obstacles. Training geometries consist of a unit square with four obstacles, one in each quadrant. The learning task is to predict the steady velocity and pressure fields.

### G.6 Augmented 2D Steady-State Navier–Stokes (NS2d-c++)

To evaluate extrapolation beyond the training geometry distribution, we introduce NS2d-c++, an augmented version of NS2d-c. While training is restricted to domains with circular obstacles, evaluation includes domains with increased obstacle count, non-circular obstacles, extended domain lengths, and angled step geometries. The governing equations and boundary conditions are unchanged in each geometry. This dataset isolates geometric generalization under fixed steady Navier–Stokes physics.

Meshes were generated in Gmsh (Geuzaine & Remacle, 2009) by randomly sampling the parametric variables (obstacle type, number, size, locations, domain length, and angle ramp). Solutions were generated in COMSOL 6.4 for steady Navier–Stokes flow with no-slip boundary conditions on the walls, a prescribed inlet velocity, and an outlet boundary condition imposing both a pressure and normal flow. The inlet velocity was prescribed as $\mathbf{u} = [4Uy(H - y)/H^2,\ 0]$, where $U = 10^{-4}\,\mathrm{m/s}$ and $H = 1\,\mathrm{m}$. We model the fluid as air. For each geometry, we provide coarse, medium, and fine levels of mesh refinement. This data will be publicly available as a medium-complexity test bed for geometry generalization in neural PDE surrogates.

## H Additional Results

### H.1 Computational cost

Geo-NeW requires a reduced nonlinear solve during inference and implicit differentiation through this solve during training. We quantify this additional cost against the inducing-point transformer baseline with the same encoder architecture in Tables 7–8 and Figure 12. Across encoder depths, Geo-NeW incurs an approximately constant additional inference cost of (3.8)–(3.9) ms relative to direct prediction, and an additional (2.6)–(3.1) s per training epoch. This cost reflects the tradeoff introduced by solving a structure-preserving reduced finite element system rather than directly regressing the solution field. For context, inference remains substantially faster than the conventional finite element solve used to generate the NS2d-c++ data: the COMSOL CPU implementation requires approximately (640) ms per sample, compared with (5.2)–(7.9) ms for Geo-NeW on a GPU. Runtime therefore increases relative to direct neural surrogates while remaining compatible with real-time evaluation and substantially faster than the reference simulator.

### H.2 Learned basis

We provide an example of the full-multifield basis functions for the NS2d-c++ dataset on a random sample in Figure 13.

Table 7. Inference time (ms) vs. encoder depth $N$.

| Method | $N = 2$ | $N = 4$ | $N = 6$ |
|---|---|---|---|
| FEM (CPU) | | 640 | |
| Baseline | 1.4 | 2.8 | 4.0 |
| Geo-NeW | 5.2 | 6.6 | 7.9 |
| $\Delta$ | 3.8 | 3.8 | 3.9 |

Table 8. Training time (s/epoch) vs. encoder depth $N$.

| Method | $N = 2$ | $N = 4$ | $N = 6$ |
|---|---|---|---|
| Baseline | 1.3 | 1.6 | 1.9 |
| Geo-NeW | 3.9 | 4.5 | 5.0 |
| $\Delta$ | 2.6 | 2.9 | 3.1 |

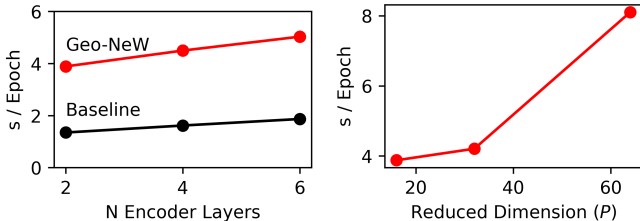

*Figure 12.* We evaluate the training runtime of our Geo-NeW model over various encoder sizes and reduced dimension, highlighting the tradeoff between expressivity and runtime, and modest increase cost from the structure preserving model compared to a baseline with an identical encoding model.

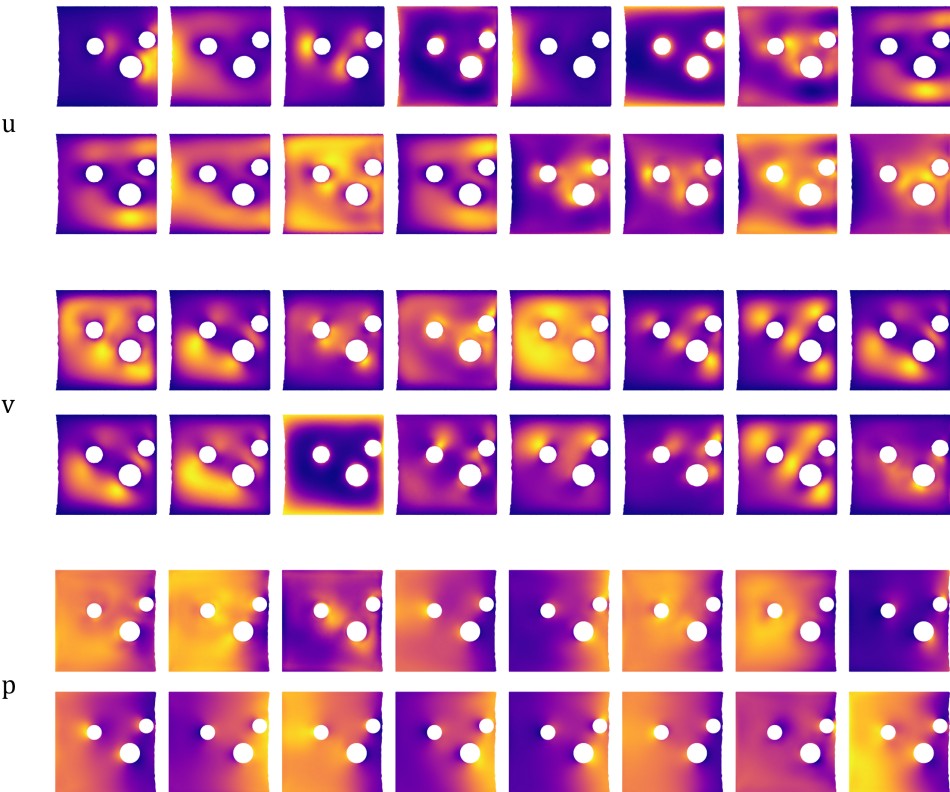

*Figure 13.* Multi-field learned basis functions for a sample from our NS2d-c++ dataset, showing the adaptation to fit the input geometry.

# I Poly-Poisson, NS2d-c, NS2d-c++ Baselines

## I.1 GNOT, Transolver, Linear-Attention

All transformer-based baselines were implemented as encoder-only models with 4 encoder layers, hidden size 128, and 4 attention heads. GNOT (Hao et al., 2023) was constructed using its proposed normalized self-attention layer and geometric gating mixture-of-experts (MoE) feed-forward module with 4 experts. For Transolver (Wu et al., 2024), we use its Physics-Attention mechanism, which learns slices that group mesh points into physics-aware tokens and performs attention over these tokens; in our configuration, we set the number of learnable slices to 64. Finally, for the linear-attention baseline, we replace softmax self-attention with the kernel/feature-map formulation of linearized attention (Katharopoulos et al., 2020), which reorders computation to reduce attention time and memory complexity from quadratic to linear in sequence length. All baselines were trained for 500,000 steps with randomly sampled minibatches (batch size 4), using the Muon optimizer (Jordan et al., 2024) and a warm-up exponential-decay learning-rate schedule.

