# OpenReview forum: "Structure-Preserving Learning Improves Geometry Generalization in Neural PDEs"
_ICML.cc/2026/Conference — ICML 2026 regular_

### Official Review · Reviewer_g58q · 2026-03-10

**Soundness:** 3
**Presentation:** 3
**Significance:** 3
**Originality:** 3
**Overall Recommendation:** 5
**Confidence:** 3

**Summary:**

A modification of the finite element method is considered, proposing the use of Neural Whitney Forms to reduce the finite element space. The method takes geometry into account and satisfies conservation laws and boundary conditions. Comparison with existing models showed superior or comparable results for solving a range of PDEs. Implementation details and code for result reproduction are also provided.

**Compliance With Llm Reviewing Policy:**

Affirmed.

**Final Justification:**

When evaluating the paper, weaknesses were identified in the organization of the paper, as well as in the comparison of the presented method's performance with both state-of-the-art and traditional solutions in the field. Additionally, the issue of OOD in the context of acceptable error margins was not sufficiently discussed. In their rebuttal, the authors provided new data and expanded the discussion of the OOD issue, which addressed these shortcomings. Based on the rebuttal, I am raising the "Soundness" score to 3. I recommend accepting the paper and am raising the overall score to 5.

**Key Questions For Authors:**

1. The results presented in Tables 1 and 2 do not show the superiority of the proposed approach in all benchmarks. Could the authors provide intuition about the conditions under which their method is preferable? A comparison of computation times would be particularly interesting.
2. Although the method shows better results for OOD, it remains unclear how usable these results are. Could the authors provide intuition about what error level would need to be achieved for an OOD problem to be considered solved?
3. Were computation time measurements conducted for traditional FEM? While the advantage presented over FEM might seem obvious, specific data on this point would strengthen confidence in the results.

**Limitations:**

yes

**Strengths And Weaknesses:**

Strengths:

1. A well-structured and comprehensive description of the method.
2. A code repository is provided for result reproduction.
3. The model is capable of working with arbitrary geometries.

Weaknesses:

See Questions.

---

> ### Author Rebuttal · Authors · 2026-03-31
>
> # Response to reviewer g58q
>
> We thank the reviewer for their constructive comments and have addressed each point below.
>
> ### **Q1.1** *“The results presented in Tables 1 and 2 do not show the superiority of the proposed approach in all benchmarks. Could the authors provide intuition about the conditions under which their method is preferable?"*
>
> As noted in other reviews, we observed that the initial NACA result was out of line with the other benchmarks, and upon further investigation identified a misspecified boundary condition in that setup (see response elsewhere). This correction reduces the error from 1.5% to 0.8%, bringing performance closer to transformer-based baselines, though still not the best-performing method. More broadly, our goal is not to uniformly outperform all baselines, but to achieve competitive in-distribution accuracy while introducing additional structure that improves generalization.
>
> Geo-NeW is most beneficial when conservation, boundary conditions, and geometry strongly influence the solution and extrapolation is required. Empirically, we observe that this advantage is most apparent in more challenging or out-of-distribution geometry settings, where Geo-NeW provides improved accuracy relative to baselines, while maintaining competitive performance on standard benchmarks.
>
> ### **Q1.2** *“A comparison of computation times would be particularly interesting.”*
>
> We agree this is an important point. We have added timing comparisons in the revised manuscript. Geo-NeW is approximately 2x slower than comparable transformer-based models for training and incurs a nearly constant 4 ms additional cost in inference due to the nonlinear solve. These timing comparisons were made on the NS2D-c dataset. However, it remains drastically faster than conventional FEM solvers for steady-state problems (see more detail in response to question 3).
>
>
> ### **Q2** *“Although the method shows better results for OOD, it remains unclear how usable these results are. Could the authors provide intuition about what error level would need to be achieved for an OOD problem to be considered solved?”*
>
> We agree performance for OOD should be clarified. Acceptable error thresholds vary by application. For engineering, errors on the order of 1–10% are commonly reported for reduced-order models and operator learning methods, with tighter tolerances required for high-precision applications and looser tolerances typically sufficient for design tasks.
>
> In addition to the detailed breakdowns in Figures 1 and 6, we better contextualize our results by introducing a “mild OOD” evaluation, restricting to geometries with L=1 (within the training range) while varying obstacle configuration and ramp angle. In this setting, Geo-NeW achieves ~8% normalized error compared to 13–15% for baselines, placing it within the range often considered usable for design-oriented tasks [ref 1, ref 2].
>
> We avoid claiming that the OOD problem is fully solved, as all methods degrade under distribution shift. However, these results demonstrate that Geo-NeW achieves meaningful accuracy in moderate extrapolation regimes, rather than only relative improvement in extreme settings.
>
> ### **Q3** *“Were computation time measurements conducted for traditional FEM? While the advantage presented over FEM might seem obvious, specific data on this point would strengthen confidence in the results.”*
>
> We agree this is an important comparison and have included FEM timing results alongside Geo-NeW and a transformer baseline. In our setup, steady-state Navier–Stokes simulations require on the order of ~1 s with FEM, compared to <10 ms for Geo-NeW, corresponding to a ~100x speedup, while Geo-NeW remains only ~4 ms slower than the transformer baseline at inference.
>
> While absolute timings depend on implementation, we observed similar orders of magnitude using both COMSOL and scikit-fem for moderately sized meshes (~12k nodes). Operating in a reduced space makes cost largely independent of mesh resolution.
>
> **Inference time (ms) vs. encoder depth $N$**
>
> | Method        | N=2 | N=4 | N=6 |
> |---------------|-----|-----|-----|
> | **FEM (CPU)** | 640 | 640 | 640 |
> | Baseline      | 1.4 | 2.8 | 4.0 |
> | Geo-NeW       | 5.2 | 6.6 | 7.9 |
> | **Δ (Geo-NeW − Baseline)** | 3.8 | 3.8 | 3.9 |
>
> More broadly, an advantage of this formulation is that it retains compatibility with the finite element framework, e.g., exact boundary condition enforcement, conservation structure, and integration with existing analysis tools such as uncertainty quantification. These guarantees and interoperability constraints are often cited as important for adoption in engineering workflows [ref 3].
>
>
> ### References
>
> [ref 1] Hong, S., et al. (2025). Journal of Mechanical Design, 147(4), 041703.
>
> [ref 2] Shukla, K., et al. (2023). arXiv preprint arXiv:2302.00807.
>
> [ref 3] National Academies of Sciences, Engineering, and Medicine. (2025). Washington, DC: The National Academies Press.

---

> > ### Author Rebuttal · Reviewer_g58q · 2026-04-03
> >
> > The author’s responses fully address the questions raised in the review. I am raising the score to 5 (accept).

---

> > > ### Author Response · Authors · 2026-04-05
> > >
> > > We are glad we were able to address all questions, and thank the reviewer for their careful evaluation, which has helped clarify the work.
> > >
> > > We would like to kindly remind the reviewer to please update the official score as well to ensure accurate evaluation.

---

### Official Review · Reviewer_ZauW · 2026-03-12

**Soundness:** 4
**Presentation:** 4
**Significance:** 3
**Originality:** 4
**Overall Recommendation:** 6
**Confidence:** 4

**Summary:**

This paper considers an important problem for learning PDEs: how to generalize across different geometries. The proposed approach is grounded in the finite element context (i.e., learning  a reduced-order finite element operator whose zero-set), resulting in "intrinsic" physical (or rather mathematical) behavior of the trained model.

**Compliance With Llm Reviewing Policy:**

Affirmed.

**Key Questions For Authors:**

None

**Limitations:**

yes

**Strengths And Weaknesses:**

The work is well-presented and seems (as far as I can judge) technically sound. The Figures and diagrams are very helpful for catching the idea of the paper -- well done. Overall, this has a high significance: learning PDEs in general is a "hot" topic, but the number of efforts for generalizing to arbitrary boundary conditions are still rather limited. I perceive the idea to focus on learning the FE operator (and not the map from input to solution itself)  to have a high degree of originality.
The only drawback is the limitation to steady-state PDEs.

---

### Official Review · Reviewer_xVqS · 2026-03-13

**Soundness:** 2
**Presentation:** 3
**Significance:** 2
**Originality:** 3
**Overall Recommendation:** 5
**Confidence:** 3

**Summary:**

This paper studies geometry generalization for neural PDE surrogates. The authors propose Geo-NeW, a geometry-conditioned implicit solver that learns reduced finite-element spaces and a learned nonlinear operator on top of mesh inputs, rather than directly regressing the solution field. A central idea is to preserve Whitney-form structure so that conservation properties and Dirichlet boundary conditions are enforced by construction. The model uses geometry-aware features such as heat-kernel signatures, harmonic coordinates, and signed-distance information, and is evaluated on several standard steady-state PDE benchmarks as well as custom out-of-distribution geometry settings. Overall, the paper positions itself as a structure-preserving alternative to more conventional neural operator or transformer-style surrogates for PDEs on variable geometries.

**Compliance With Llm Reviewing Policy:**

Affirmed.

**Final Justification:**

The author resolved my concerns.

**Key Questions For Authors:**

1. Could the authors add FNO results to Table 2? Since these are Geo-FNO standard benchmarks, this would materially improve comparability. If Geo-NeW remains competitive against FNO there, that would increase my confidence in the empirical case.

2. Could the authors better contextualize NS2d-c++ OOD with either milder OOD settings, alternative error metrics, or a more detailed per-subsetting breakdown? My evaluation would improve if the paper showed that the method remains meaningfully accurate in at least some nontrivial extrapolation regimes, rather than only relatively better in a regime where all methods have very large error.

3. Could the authors move the implied-boundary-condition caveat for NACA and Elasticity into the main text and explain more clearly how this affects comparability? A clearer discussion here would improve my confidence in the standard benchmark claims.

4. Could the authors clarify how far they believe the current experiments support the broader geometry-generalization claims beyond 2D steady-state settings? A more calibrated discussion of scope would make the contribution easier to place and evaluate fairly.

**Limitations:**

Yes

**Strengths And Weaknesses:**

Soundness. From a soundness perspective, the paper is technically well motivated: the method is posed as a PDE-constrained implicit solve, uses a reduced finite-element formulation, and explicitly enforces Dirichlet boundary conditions rather than treating them only as a soft objective. I also appreciate that the evaluation is not limited to a single dataset and includes both standard steady-state geometry benchmarks and more challenging extrapolation settings. My concern is that the empirical support is somewhat uneven. In Table 2, Geo-NeW is very strong on Elasticity and Pipe, but clearly weaker on NACA, and the NACA/Elasticity numbers are marked as using implied boundary conditions, which makes direct comparison less clean. Since the paper states that these standard tasks come from Geo-FNO, I also think Table 2 should include an FNO result for the same standard prediction setting. Finally, the NS2d-c++ OOD result is directionally encouraging but not fully convincing as practical accuracy, because all methods have very large relative errors in that regime: the baselines are around 83–92 while Geo-NeW is still 42.2, and the paper itself notes that coordinate-based baselines exceed ε>30 under these extrapolation settings. Overall, I view the work as technically serious, but the empirical case would be stronger with cleaner benchmark comparability and a more cautious interpretation of the OOD numbers.

Presentation. The paper is generally readable and the overall narrative is coherent: the motivation for moving beyond direct solution regression is clear, and the connection between geometry encoding, reduced FE spaces, and implicit solution is explained reasonably well. The work is also positioned in a sensible context relative to neural operators and geometry-general PDE learning. My main presentation concern is that some of the most important experimental qualifications are too easy to miss. In particular, the implied-boundary-condition caveat for Table 2 appears only as a dagger note, and the standard-benchmark comparison would be easier to interpret if the paper more explicitly explained why those settings differ from the original PDE setups and why FNO is not shown despite the benchmarks being from Geo-FNO. I also think the severity of the NS2d-c++ extrapolation regime should be stated more prominently in the main discussion, because otherwise readers may over-interpret the headline relative improvement. So I find the paper reasonably well written, but I would encourage the authors to surface key evaluation caveats more clearly in the main text rather than leaving them implicit in tables or appendix details.

Significance. I do think the paper addresses an important problem. Geometry generalization is a real bottleneck for scientific machine learning, and a method that combines geometric conditioning with exact boundary handling and conservation-aware structure could matter for future research on neural PDE solvers and design-oriented surrogates. In that sense, the paper has meaningful potential significance even if the contribution is somewhat specialized. My hesitation is mostly about demonstrated scope: the experiments are currently confined to steady-state 2D problems, and the conclusion frames 3D as a natural extension rather than something established here. Combined with the fact that the strongest extrapolation evidence comes from a custom NS2d-c++ benchmark where absolute errors remain very large, I see this work more as a promising step in an important direction than as a decisive advance in practical capability today.

Originality. On originality, I am positive. This does not read as a minor variant of a standard neural operator; rather, the novelty comes from combining geometry-conditioned reduced FE spaces, FEEC/Whitney-form structure, intrinsic geometry descriptors such as HKS/HC/SDF, and an implicit PDE-constrained solve in a way that is clearly designed around geometry generalization. I think that combination is thoughtful and well motivated, and the paper does articulate why these pieces fit together. At the same time, the originality is strongest in the formulation itself rather than in overwhelmingly decisive empirical gains, so stronger standardized comparisons would help justify the broader claims. Overall, I see the paper as methodologically original and interesting, even though my overall stance remains borderline negative because the evaluation does not yet fully match the ambition of the underlying idea.

---

> ### Author Rebuttal · Authors · 2026-03-31
>
> # Response to reviewer xVqS:
>
> We thank the reviewer for their thorough and constructive remarks.
>
> ### **Q1** *Addition of Geo-FNO*
>
> We agree that this is an important comparison and have added the Geo-FNO [ref 1] results to Table 2 in the revised manuscript. Additionally, we have corrected a misspecified boundary condition for the NACA experiment which results in improved performance (1.5% -> 0.8%), which is more in line with transformer baselines. Geo-NeW remains competitive across benchmarks.
>
> **Table 1: Prediction error across Standard PDE datasets**
>
> | Model | NACA ↓ | Elasticity ↓ | Pipe ↓ |
> |---|---|---|---|
> | U-Net | 0.79 | 2.35 | 0.65 |
> | DeepONet | 3.85 | 9.65 | 0.97 |
> | Geo-FNO | 1.38 | 0.67 | 0.74 |
> | Galerkin | 1.18 | 2.40 | 0.98 |
> | GNOT | 0.76 | 0.86 | 0.47 |
> | Transolver | 0.53 | 0.64 | 0.46 |
> | LaMO | **0.41** | 0.50 | 0.38 |
> | **Ours (Geo-NeW)** | 0.82† | **0.35†** | **0.11** |
>
> † modeling derived quantity
>
>
> ### **Q2** *“Could the authors better contextualize NS2d-c++ OOD ...”*
>
> We introduced a “NS2d-c++ OOD (mild)” evaluation column to the results in Table 1, which restricts the evaluation to domains with a length of 1 (i.e. the same as the training examples). In this setting the baselines approaches provide 13-15% error and Geo-NeW has 7.8%, resulting in a 44% relative improvement over the second best (linear attention). While the threshold for a “qualitatively useful” error varies by application, 5-10% have been considered sufficient for design in similar domains [ref 2, ref 3]. In addition to the relative error breakdown in Figures 1 and 6 for variable theta, L, and number of obstacles. This shows consistent improvement across extrapolation regimes although without eliminating distribution shift error entirely.
>
> | Model | NS2d-c++ OOD | OOD (mild) |
> |---|---|---|
> | GNOT | 83.1 | 14.1 |
> | Transolver | 91.4 | 15.1 |
> | Linear Attn | 91.8 | 13.1 |
> | Inducing Point | -- | -- |
> | **Ours (Geo-NeW)** | **42.2** | **7.87** |
>
>
> ### **Q3** *Clarification of implied boundary conditions*
>
> We thank the reviewer for the feedback and agree this should be clarified in the main text. The description of the “implied boundary conditions” was not clearly present in the original draft and has been revised and moved to section 4.1.
>
> The standard “NACA” and “Elasticity” datasets are formulated for pointwise prediction of derived quantities (Mach number and von Mises stress), rather than the primary state variables of the governing PDE. To maintain comparability, Geo-NeW in these settings learns an effective PDE posed directly on the reported field, rather than recovering the original underlying PDE.
>
> We impose boundary conditions on these fields to define a well-posed boundary value problem in this representation. Specifically, we apply Dirichlet boundary conditions on the modeled fields at the inlet (NACA) and the clamped boundary (Elasticity). The original “implied boundary condition” designation did not clearly capture this distinction; we have updated it to “modeling derived quantity” and have clarified this interpretation in the main text under Section 4.1. We note that this is the same target quantity used in the baselines.
>
> ### **Q4** *Clarification of scope*
>
> Concretely, we demonstrate improved generalization to out-of-distribution geometries for canonical 2D steady-state PDEs (including Navier–Stokes). We attribute this to the proposed formulation, which combines: (1) learnable reduced finite element models, (2) a geometry-conditioned reduced FE space, and (3) a stable flux parameterization for consistent training.
>
> The formulation is defined natively from mesh-derived operators and is therefore not dimension-specific. Extending to 3D primarily introduces practical challenges related to scaling and conditioning on large meshes rather than changes to the core model. While the formulation in [ref 4] supports time-dependent problems in abstract, we view validation in these settings as an important direction for future work, as they introduce additional challenges such as temporal stability, long-horizon error accumulation, and data requirements for time-resolved dynamics. Simultaneously, the flux-based PDE structure and FE discretization provide natural benefits for conservative time integration and stable autoregressive rollouts, which should be explored in future work.
>
> ### **Additional Points 1.** *Uneven performance between baselines.*
>
> We agree with this observation, as noted in responses to the other reviews, we identified a misspecified boundary condition for the NACA example. Due to space limits we direct to the response to reviewer Ya5s for more details.
>
> ### References:
> [ref 1] Li, Z., et al. (2023). JMLR, 24(388), 1-26.
>
> [ref 2] Hong, S., et al. (2025). Journal of Mechanical Design, 147(4), 041703.
>
> [ref 3] Shukla, K., et al. (2023). arXiv preprint arXiv:2302.00807.
>
> [ref 4] Kinch, B., et al. (2025). arXiv preprint arXiv:2508.06981.

---

> > ### Author Rebuttal · Reviewer_xVqS · 2026-04-03
> >
> > My questions is resolved. Thanks very much.

---

### Official Review · Reviewer_Ya5s · 2026-03-13

**Soundness:** 3
**Presentation:** 3
**Significance:** 3
**Originality:** 3
**Overall Recommendation:** 4
**Confidence:** 3

**Summary:**

This paper studies PDE foundation models which generalizes to unseen geometry. Geometry is encoded by combining HKS, HC and SDF. The representation method guarantees translation, rotation and discretization invaraince. By implementing Finite element exterior calculus, conservation laws and boundary conditions are naturally guaranteed. Training is achieved by implicit differentiation through a PDE solver. The proposed method is compared with DeepONet, U-Net, Galerkin, GNOT, Transolver and LaMO, on benchmarks including NACA, Elasticity, Pipe and NS2d-c.

**Compliance With Llm Reviewing Policy:**

Affirmed.

**Final Justification:**

The authors have provided detailed information during rebuttal for all reviewers. Overall I will keep my already positive rating.

**Key Questions For Authors:**

- How computationally efficient is the proposed model compared to baseline methods?
- On the NACA dataset, the proposed model performs worse than all previous foundation models, while it performs the best on Elasticity and Pipe. Can you provide some explanations?

**Limitations:**

Limitations are not explicitly discussed in the main text or appendix. Potential limitations include neural surrogate PDE solvers are not suitable for precision-sensitive scenarios.

**Strengths And Weaknesses:**

Strengths:
- The proposed model guarantees a wide range of priors, including translation, rotation and discretization invaraince, as well as conservation laws and boundary conditions.

Weaknesses:
- The proposed method will be very expensive to train.
- There are potential expressivity limits due to stability constraints.

Minor Issues:
- Some abbreviations are first used before their full forms, such as HKS, HC and SDF.

---

> ### Author Rebuttal · Authors · 2026-03-31
>
> # Response to reviewer Ya5s
>
> We thank the reviewer for their constructive review. We have addressed each question below.
>
> ### **P1** *“The proposed method will be very expensive to train.”*
>
> Our approach requires differentiable FE-solves, which does incur additional cost compared to direct evaluation. However, in practice this additional cost is modest: Geo-NeW is approximately 2x slower per training epoch than the comparable inducing-point transformer model and remains efficient for inference (see response to Q1). We find this to be a reasonable tradeoff for enabling extrapolation to new geometries and incorporating benefits from finite element structure (e.g., exact boundary conditions, conservation laws, interoperability with conventional numerical methods).
>
> **Training time (s / epoch) vs. encoder depth $N$**
>
> | Method        | N=2 | N=4 | N=6 |
> |---------------|-----|-----|-----|
> | Baseline      | 1.3 | 1.6 | 1.9 |
> | Geo-NeW       | 3.9 | 4.5 | 5.0 |
> | Δ             | 2.6 | 2.9 | 3.1 |
>
> This efficiency is enabled by particular design choices: (1) we utilize implicit differentiation and the well-posedness of the learned FE system to backpropagate without unrolling the nonlinear solve in training, (2) the solve takes place in the reduced FE space which has a fixed dimension P << N (i.e. P=16), (3) our Lipschitz constrained flux model results in solves which converge in few iterations (Figure 11 in appendix). Overall training remains practical for the problem sizes considered.
>
>
> ### **P2** *“There are potential expressivity limits due to stability constraints.”*
>
> We agree that imposing stability constraints restricts the class of admissible flux functions. In our formulation, the Lipschitz constraint ensures that the nonlinear system remains well-conditioned and solvable across geometries, which enables the efficient training via implicit differentiation. Comparisons to baseline regression approaches do not indicate an empirical degradation in performance from these constraints as Geo-NeW achieves competitive or improved accuracy across several benchmarks. This suggests that the restricted function class remains sufficiently expressive for the problems considered.
>
> ### **P3** *“Limitations are not explicitly discussed in the main text or appendix. Potential limitations include neural surrogate PDE solvers are not suitable for precision-sensitive scenarios.”*
>
> We have added a limitations section in the appendix. In addition to the reviewer’s point, we note that our approach incurs an additional 4ms inference time and 2x training time computational cost and requires a mesh and boundary condition specification, which are expected in scientific computing.
>
> ### **Q1** *“How computationally efficient is the proposed model compared to baseline methods?”*
>
> Geo-NeW is approximately 2x slower than comparable transformer-based models for training and incurs a nearly constant 4 ms additional cost in inference (independent of encoder size), for the steady state Navier-Stokes examples. We have expanded and clarified these timing results in the revised appendix. We also note in Figure 11 (b) in the appendix that these timing results are dependent on the number of solve iterations needed; for the NS2D-c example this is 3.0 iterations on average. We additionally maintain orders of magnitude speedup over conventional FEM solvers, while maintaining the essential finite element structure. FEM timings are based on a CPU implementation of steady state Navier-Stokes in COMSOL.
>
> The current implementation is not as optimized as standard transformer architectures; there may be opportunities for future improvements.
>
> **Inference time (ms) vs. encoder depth $N$**
>
> | Method        | N=2 | N=4 | N=6 |
> |---------------|-----|-----|-----|
> | **FEM (CPU)** | 640 | 640 | 640 |
> | Baseline      | 1.4 | 2.8 | 4.0 |
> | Geo-NeW       | 5.2 | 6.6 | 7.9 |
> | **Δ (Geo-NeW − Baseline)** | 3.8 | 3.8 | 3.9 |
>
>
> ### **Q2** *Explain variance in performance*
>
> We also observed that the initial NACA results were out of line with the other benchmarks, upon further investigation we identified a misspecified boundary condition in the NACA setup. This correction reduces the normalized error from 1.5% to 0.8%, bringing performance closer to transformer-based baselines.
>
> We aim to achieve competitive in-distribution performance while introducing benefits from finite element structure and improving out-of-distribution extrapolation. In several cases, this reformulation provides useful inductive biases and leads to improved accuracy. At the same time, since Geo-NeW learns a representation of the underlying physics rather than directly regressing the solution, we expect performance to vary across different physical settings (e.g. hyberbollic vs elliptic PDEs), rather than exhibiting uniform gains across all benchmarks.

---

> > ### Author Rebuttal · Reviewer_Ya5s · 2026-04-01
> >
> > All my concerns, including training and inference efficiency, expressivity limitations, and performance analysis on NACA dataset, have been responded.

---

> > > ### Author Response · Authors · 2026-04-04
> > >
> > > We are glad we were able to address your concerns and appreciate your careful evaluation.
> > >
> > > Please let us know if there are any remaining points we can clarify or improve as you finalize your evaluation.

---

### Decision · Program_Chairs · 2026-04-30

**Decision:**

Accept (regular)

**Comment:**

All four reviewers are in favor of acceptance. They found the implicit PDE-constrainted solve approach technically novel, the paper well written and they appreciated the generalization across geometries and the large speed over FEM. The limitation to 2D steady-state PDEs is acknowledged but not a deal-breaker for a first demonstration. I recommend acceptance.